# Co-translational assembly of mammalian nuclear multisubunit complexes

Ivanka Kamenova [1,2,3,4,7], Pooja Mukherjee [1,2,3,4,7], Sascha Conic[1,2,3,4], Florian Mueller [5], Farrah El-Saafin[1,2,3,4], Paul Bardot[1,2,3,4], Jean-Marie Garnier[1,2,3,4], Doulaye Dembele [1,2,3,4], Simona Capponi[6], H.T.Marc Timmers[6], Stéphane D. Vincent [1,2,3,4] & László Tora [1,2,3,4]

Cells dedicate significant energy to build proteins often organized in multiprotein assemblies with tightly regulated stoichiometries. As genes encoding subunits assembling in a multi-subunit complex are dispersed in the genome of eukaryotes, it is unclear how these protein complexes assemble. Here, we show that mammalian nuclear transcription complexes (TFIID, TREX-2 and SAGA) composed of a large number of subunits, but lacking precise architectural details are built co-translationally. We demonstrate that dimerization domains and their positions in the interacting subunits determine the co-translational assembly pathway (simultaneous or sequential). The lack of co-translational interaction can lead to degradation of the partner protein. Thus, protein synthesis and complex assembly are linked in building mammalian multisubunit complexes, suggesting that co-translational assembly is a general principle in mammalian cells to avoid non-specific interactions and protein aggregation. These findings will also advance structural biology by defining endogenous co-translational building blocks in the architecture of multisubunit complexes.

[1] Institut de Génétique et de Biologie Moléculaire et Cellulaire, 67404 Illkirch, France. [2] Centre National de la Recherche Scientifique (UMR7104), 67404 Illkirch, France. [3] Institut National de la Santé et de la Recherche Médicale (U1258), 67404 Illkirch, France. [4] Université de Strasbourg, Illkirch 67404, France. [5] Computational Imaging & Modeling Unit, Institut Pasteur, Département Biologie Cellulaire et Infections, 25-28 rue du Docteur Roux, 75015 Paris, France. [6] German Cancer Consortium (DKTK) partner site Freiburg, German Cancer Research, Center (DKFZ) and Department of Urology, Medical Center-University of Freiburg, 79106 Freiburg, Germany. [7] These authors contributed equally: Ivanka Kamenova, Pooja Mukherjee. Correspondence and requests for materials should be addressed to L.T. (email: laszlo@igbmc.fr)

Often proteins do not act alone, instead they function as components of large multisubunit complexes in a cell. To better understand cellular functions, investigating the precise mechanism that guide the formation of these multisubunit assemblies is of key importance. A cell uses hundreds of different protein complexes that vary with respect to their complexity. Some complexes require the association of multiple copies of the same subunit, while others are constituted of many different subunits. The latter group includes many transcription regulatory and chromatin remodelling complexes (see below). In order to achieve the efficient formation of protein complexes in eukaryotes, the genes coding for all the subunits (dispersed in the eukaryotic genome) have to be transcribed in the nucleus, their corresponding mRNAs transported to the cytoplasm, translated into proteins, and the formation of correct interactions among the subunits must be orchestrated. A polysome is a cluster of ribosomes acting on a single mRNA to translate its information into polypeptides. Appropriate translation-based mechanisms may exist in the cell to regulate the interactions between specific subunits in order to avoid incorrect non-specific interactions or subunit aggregations in the absence of the correct partner. Currently, it is not well understood how functional subunit interactions are regulated in eukaryotic cells. Protein complex formation is often studied in vitro using purified subunits, assuming that individually translated subunits assemble stochastically by diffusion, and thus favouring the idea that these multisubunit complexes assemble post-translationally[1]. However, in the crowded environment of an eukaryotic cell such simple diffusion-dependent models may not work, as subunits may engage in non-specific interactions or form aggregates. Recent studies in bacteria demonstrated that co-translational building of a functional protein dimer is more efficient than the post-translational assembly of its individual subunits[2,3], and also in yeast co-translation has been shown to be an efficient assembly pathway to assemble multiprotein complexes[4–8]. Consequently, two co-translational models have been put forward: (i) the simultaneous model which suggests that two polysomes in close physical proximity synthesise subunits, which interact while being translated and (ii) the sequential model implies that a mature fully translated subunit interacts co-translationally with its polysome-bound nascent interaction partner[9].

One of the key regulatory steps in the expression of mRNAs is transcription initiation. Co-activators act together to establish a chromatin structure favourable for transcription by facilitating the formation of the preinitiation complex (PIC). PIC is comprised of RNA polymerase II (Pol II) and general transcription factors (GTFs). Many GTFs and co-activators are multisubunit complexes, in which individual subunits are organised into several distinct modules carrying out specific functions. In mammalian cells the TFIID GTF nucleates the assembly of the Pol II preinitiation complex on all protein-coding gene promoters [refs [10,11] and references therein]. Metazoan TFIID is composed of the TATA-binding protein (TBP) and 13 TBP-associated factors (TAFs) (Fig. 1a). SAGA (Spt Ada Gcn5 Acetyltrasferase) is a multisubunit transcriptional coactivator complex, composed of 19 subunits (including a subset of TAFs), required for the transcription of all active genes in yeast[12]. Moreover, the mammalian Transcription and mRNA Export 2 complex (TREX-2) is composed of five subunits, including the subunit ENY2, which is shared with the SAGA complex[13].

The majority of TAFs dimerise via their histone-fold domains (HFDs), which are structurally homologous to histone pairs. In TFIID, TAFs form five HF pairs (TAF4-12, TAF6-9, TAF8-10, TAF3-10 and TAF11-13) [ref. [10] and references therein] (Fig. 1a). Importantly, individual HFD-containing TAFs cannot be expressed in a soluble form in bacteria. However, HFD-containing TAFs become soluble when co-expressed with their corresponding specific interaction partner[14], suggesting that individual HFD-containing TAFs aggregate without their specific partners.

To test how mammalian cells can avoid the aggregation of individual subunits following translation and whether co-translational interactions guide the assembly of transcription complexes, in this study, we investigate pairwise assembly of TFIID subunits between TAF8 and TAF10, TAF6 and TAF9 and TAF1 and TBP in polysome-containing mammalian cell extracts. By using a large series of complementary experiments, we show that TAF8-TAF10 and TAF1-TBP assemble co-translationally according to the sequential assembly pathway, while TAF6-TAF9 assembles co-translationally according to the simultaneous model. We also demonstrate that the ENY2 subunit assembles co-translationally with its interaction partner, GANP, in TREX-2, and with ATXN7L3 in the deubiquitination (DUB) module of SAGA. Furthermore, our experiments show that the interaction domain (ID) and the position of the ID in the given subunit solely drives the co-translational assembly in these complexes. Thus, our results uncover mechanistic principles in the understanding of co-translational control of protein complex formation in mammalian cells.

## Results

**TAF10 and TAF8 assemble co-translationally.** To test whether HFD-containing TAFs assemble co-translationally, we used a monoclonal antibody against the N-terminus of the HFD-containing TAF10 to immunoprecipitate (IP) endogenous TAF10 from human HeLa cell cytosolic polysome extracts (Fig. 1b). Protein–protein interactions between nascent proteins still associated with translating ribosomes would be revealed by enrichment of mRNAs coding for the interacting partners in the IPs. Global microarray analysis of mRNAs precipitated by the anti-TAF10 RNA IPs (RIPs) revealed enrichment of *TAF8* mRNA, suggesting that the well-characterised TAF8-10 HFD dimer[15] forms co-translationally (Fig. 1c). Anti-TAF10 RIP of cytosolic polysome extracts coupled to RT-qPCR validation confirmed our microarray results and showed strong enrichment of the *TAF8* mRNA (Fig. 1d). The absence of significant *TAF10* mRNA signal in the microarray experiments was due to poor quality and the high GC-content of the *TAF10* probe sets present on the commercial microarray. Nevertheless, RT-qPCR validation also revealed the presence of *TAF10* mRNA in the nascent anti-TAF10 RIP. Importantly, cycloheximide, which freezes translating ribosomes on the mRNA[16], stabilised the TAF10-TAF8-*TAF8* mRNA interactions, while puromycin, which causes release of nascent peptides from ribosomes[17], resulted in the loss of co-purified mRNA. Endogenous anti-TAF10 RIP-RT-qPCR from polysome extracts prepared from mouse embryonic stem cells (mESCs) gave nearly identical results, which emphasises the generality of the co-translational pathway for assembly of the mammalian TAF8-TAF10 heterodimer (Fig. 1e; Supplementary Fig. 1). Quantification of the *TAF8* mRNA in the anti-TAF10 RIP normalised to the protein IP efficiency indicated that the enrichment was between 7 and 25%, depending on the cell line and the antibody used. In contrast, to *TAF8*, mRNAs encoding other potential TAF10 dimerization partners, TAF3 and SUPT7L[18], were not enriched in the RT-qPCR validation experiments, in good agreement with the microarray analysis and indicating the specificity of the co-translational assembly of the TA8-TAF10 heterodimer (Fig. 1d, e). Together these results indicate that TAF10 protein is associated with ribosomes which are actively translating *TAF8* mRNA via the nascent TAF8 protein.

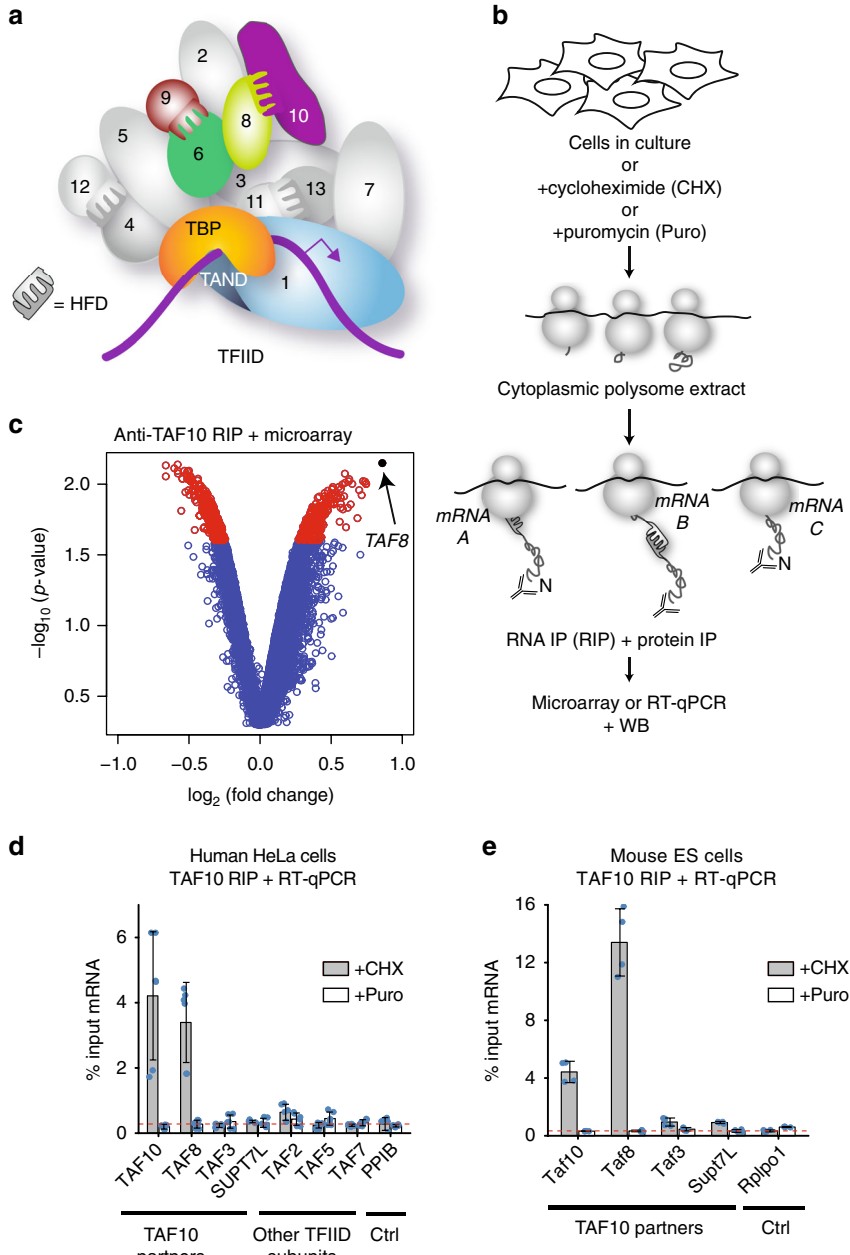

**Fig. 1** Co-translational assembly of mammalian TFIID. **a** TFIID bound to promoter DNA. TFIID is composed of TBP and 13 TAFs (indicated by numbers). Subunits analysed in this study are highlighted in colour. The histone fold domain (HFD) interactions and the TBP-TAF1 TAND domain interaction are highlighted. **b** Schematic representation of polysome RIP assay. **c** Endogenous TAF10 was immunoprecipitated from HeLa polysome-containing extract using an antibody targeting the N-terminus of the protein. The enrichment of the precipitated RNAs was assessed globally by microarray. Volcano plot depicting microarray results as log2 of the fold change of IP over a mock IP. A $p$-value cut-off $\leq 0.025$ was applied and corresponding transcripts are in red. *TAF8* transcript is highlighted in black. **d**, **e** RIP-qPCR validation of the microarray results in HeLa (**d**) and mouse ES (**e**) cells. Error bars are ±SD from three (HeLa) or two (mESC) biological replicates and two technical replicates (represented by blue dots). *Ctrl* = negative control mRNA. *PPIB* and *Rplp0* were used as unrelated control mRNAs. Source data provided as a Source Data File

**HFD drives the co-translational assembly of TAF10-TAF8.** The fact that TAF8 has its dimerization HFD at an N-terminal position, and that the TAF10 HFD is at the very C-terminus of the protein, allows the direct testing of the sequential assembly model, as TAF8 and TAF10 may be expected to only heterodimerise if the TAF10 protein is fully synthesised and freed from the ribosome. To examine the two assembly models (see Introduction) and to distinguish between the nascent and mature forms of the TAF8 and TAF10 proteins, we added FLAG-, or HA-

tags to either N- (to carry out nascent IPs) or C termini (to carry out mature IPs) of these proteins, respectively. Importantly, exogenous co-expression of N-terminally tagged TAF8 and TAF10 in HeLa cells followed by nascent anti-HA-TAF10 RIP from cytosolic polysome extracts recapitulated the findings obtained with endogenous proteins (Fig. 2a). In contrast, nascent anti-FLAG-TAF8 RIP resulted in high enrichment of its own encoding mRNA, but not that of *TAF10* (Fig. 2b). Immunoprecipitation of mature TAF10-HA protein resulted in *TAF8* mRNA,

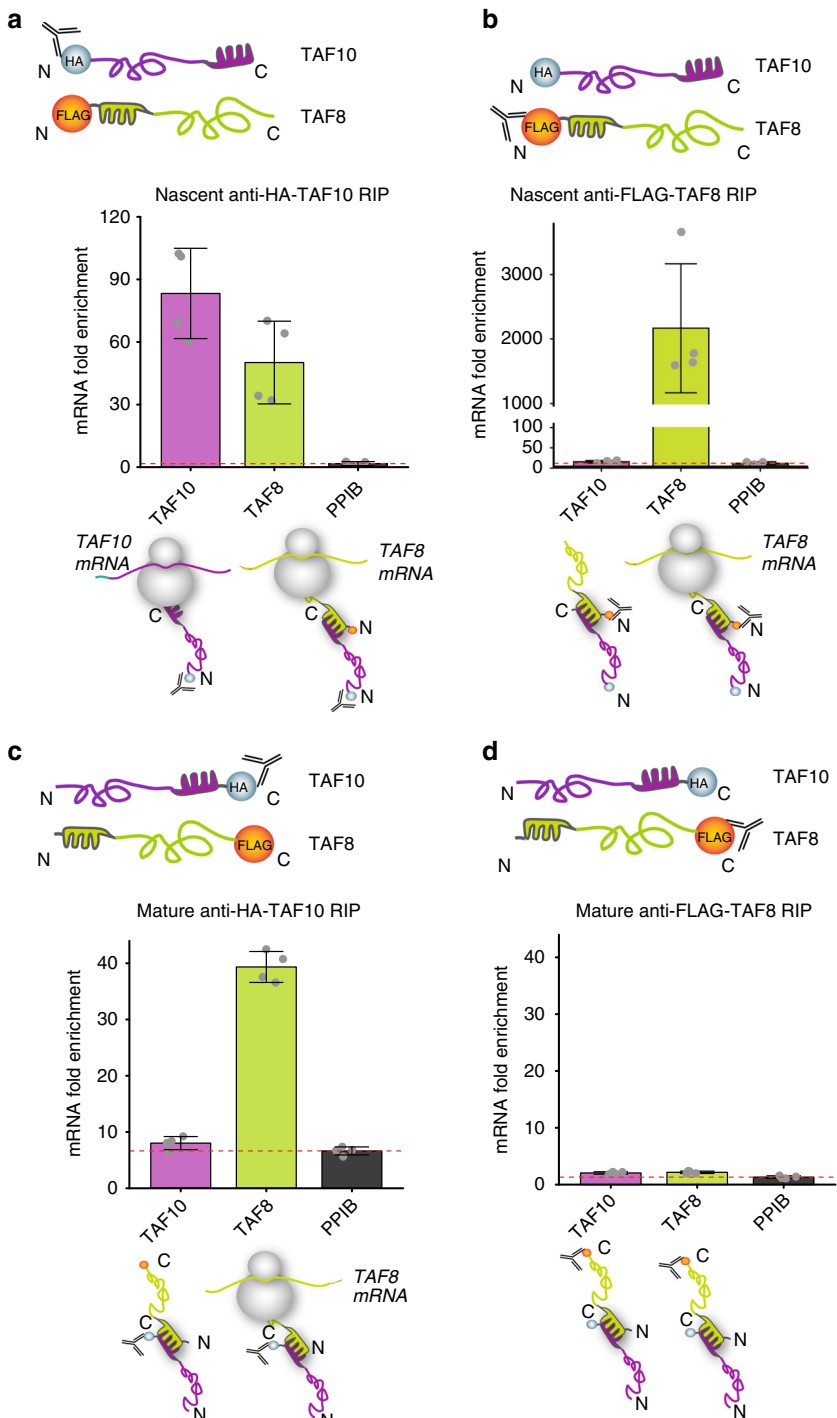

**Fig. 2** Sequential assembly of TAF10 and TAF8. N-terminal (**a**) and C-terminal (**c**) anti-HA RIP-qPCR of HeLa cell polysome extracts, co-transfected with the corresponding TAF8 and TAF10 expression plasmids (as indicated). N-terminal (**b**) and C-terminal (**d**) anti-FLAG RIP-qPCR of HeLa cell polysome extracts, co-transfected with the indicated expression plasmids. *PP*IB (**a**–**d**) was used as negative control mRNA. In panels (**a**–**d**) mRNA fold enrichment is expressed as fold change with respect to the mock IP calculated by the formula $\Delta\Delta$Cp [IP/mock] and error bars represent ±SD from two biological and two technical replicates (represented by grey dots). Source data provided as a Source Data File

but not *TAF10* mRNA enrichment (Fig. 2c), supporting the sequential co-assembly model of mature TAF10 interacting with nascent TAF8 exiting from ribosomes translating TAF8 mRNA. In addition, the mature TAF8-FLAG protein did not bring down any of the tested mRNAs (Fig. 2d). In all cases, protein partners were co-immunoprecipitated successfully (Supplementary Fig. 2).

Taken together, these results suggest that mature TAF10 binds to the polysome-bound nascent TAF8 protein, and that the respective N- (in TAF8) and C-terminal (in TAF10) HFDs are driving co-translational dimerization.

To test whether the observed co-translational TAF8-TAF10 assembly is specific to the dimerization of their HFDs, we

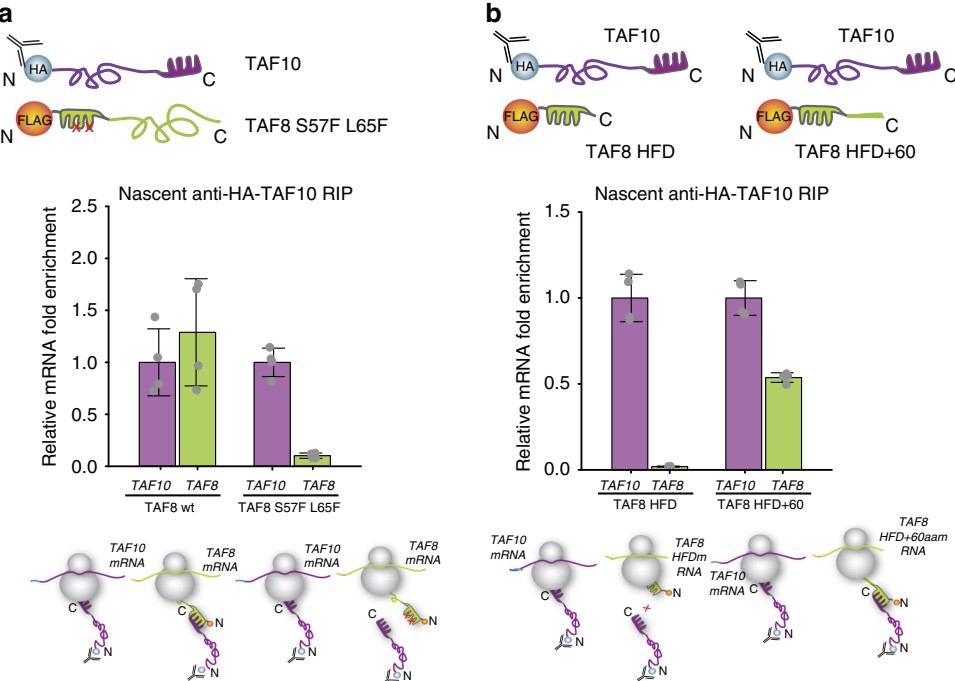

**Fig. 3** Protein-protein ID drives co-translational assembly of TAF10 and TAF8. **a** anti-HA-TAF10 RIP-qPCR of HeLa cells co-transfected with HA-TAF10 and either wild type FLAG-TAF8 (TAF8 wt) or mutant FLAG-TAF8 (TAF8 S57F L65F). **b** anti-HA-TAF10 RIP-qPCR of HeLa cells co-transfected with HA-TAF10 and either a minimal TAF8 HFD or TAF8 HFD extended with 60 amino acids (TAF8 HFD + 60aa). In panels (**a**, **b**) relative mRNA fold enrichment is expressed as fold change of *TAF8* mRNA (with respect to the mock IP calculated by the formula ΔΔCp [IP/mock]) relative to *TAF10* mRNA and error bars represent ±SD from two biological and two technical replicates (represented by grey dots). Source data provided as a Source Data File

engineered a mutation disrupting the dimerization ability of the TAF8 HFD (see Methods). Anti-TAF10 RIP from cells co-transfected with *TAF10* cDNA and mutant HFD expressing *TAF8* cDNA (mtTAF8) resulted in a nearly complete loss of the co-precipitated *TAF8* mRNA and TAF8 protein, as compared with the wild-type controls (Fig. 3a and Supplementary Fig. 3a), indicating that the dimerization of TAF8 and TAF10 through their HFDs is crucial for co-translational assembly.

Next, we tested whether the full exposure of the nascent HF interaction domain at the ribosomal exit tunnel would be necessary for co-translational assembly. The ribosome exit tunnel can accommodate up to 60 amino acids [ref. [19] and references therein]. Thus, we constructed two truncated versions of TAF8: one encoding only the TAF8 HFD that would be partially buried in the ribosome exit tunnel during translation, and a second encoding the TAF8 HFD and an additional 60 amino acids of TAF8 (TAF8 HFD + 60) that would allow the appearance of the nascent TAF8 HFD from the ribosomal tunnel. Next a TAF10 expressing plasmid was co-transfected either with TAF8 HFD, or with TAF8 HFD + 60 expressing plasmids and anti-TAF10 RIPs were carried out. Importantly, our results show that the *TAF8 HFD* mRNA is not enriched in the anti-TAF10 RIP, indicating that the minimal TAF8 HFD protein is released immediately from translating polysomes without co-translational binding to TAF10 protein. On the other hand, the *TAF8 HFD + 60* mRNA was enriched in the anti-TAF10 RIP demonstrating that the additional 60 amino acids in the longer TAF8 HFD + 60 protein kept the nascent protein anchored in polysomes allowing for co-translational interaction with TAF10 (Fig. 3b and Supplementary Fig. 3b). Together, our results indicate that TAF8-TAF10 co-translational assembly is driven by dimerization with nascent TAF8 protein upon emergence of its entire HFD from actively translating polysomes. Consequently, these results together

demonstrate the sequential co-translational assembly pathway where the fully synthesised TAF10 interacts uni-directionally with the nascent TAF8 polypeptide.

**TAF8 is prone to degradation in the absence of TAF10.** In the sequential assembly pathway, if nascent chains of a subunit cannot co-translationally interact with its partner, it may become prone to misfolding and degradation by the proteasome, but the fully translated partner should stay stable. To test this hypothesis, we used mouse embryonic stem cells (ESCs) in which either the endogenous *Taf10*, or *Taf8* genes can be conditionally knocked out[20,21]. By using these mouse ESCs we observed that the deletion of *Taf10* not only ablated *Taf10* mRNA and TAF10 protein levels, but significantly reduced both *Taf8* mRNA and TAF8 protein expression (Fig. 4a, c). These results were also confirmed in *Taf10* KO mouse embryos[20]. In contrast, the deletion of *Taf8*, decreased only its own mRNA and protein levels, without affecting the *Taf10* mRNA expression and TAF10 protein levels (Fig. 4b, d). Furthermore, in both KO mESCs other tested TFIID subunits remained unchanged[20].

Next we tested whether TAF10 re-expression would rescue TAF8 from degradation. To this end we used our *Taf10⁻ᐟ⁻:R* mouse F9 cells, where the endogenous *Taf10* alleles are inactivated and the cells are viable due to the doxycyclin (Dox) inducible expression of the human TAF10 protein[22]. In this system cells were grown for 5 days without Dox. As a result TAF10 was completely depleted and consequently endogenous TAF8 expression was also abolished (Fig. 4e), in agreement with our above mESC results. Importantly, however, when after 5 days Dox was re-added to the cells for 1 or 2 days, the neosynthesised TAF10 expression re-stabilised the expression of endogenous TAF8 as both TAF10 and TAF8 proteins could again be detected by western blot analysis (Fig. 4e).

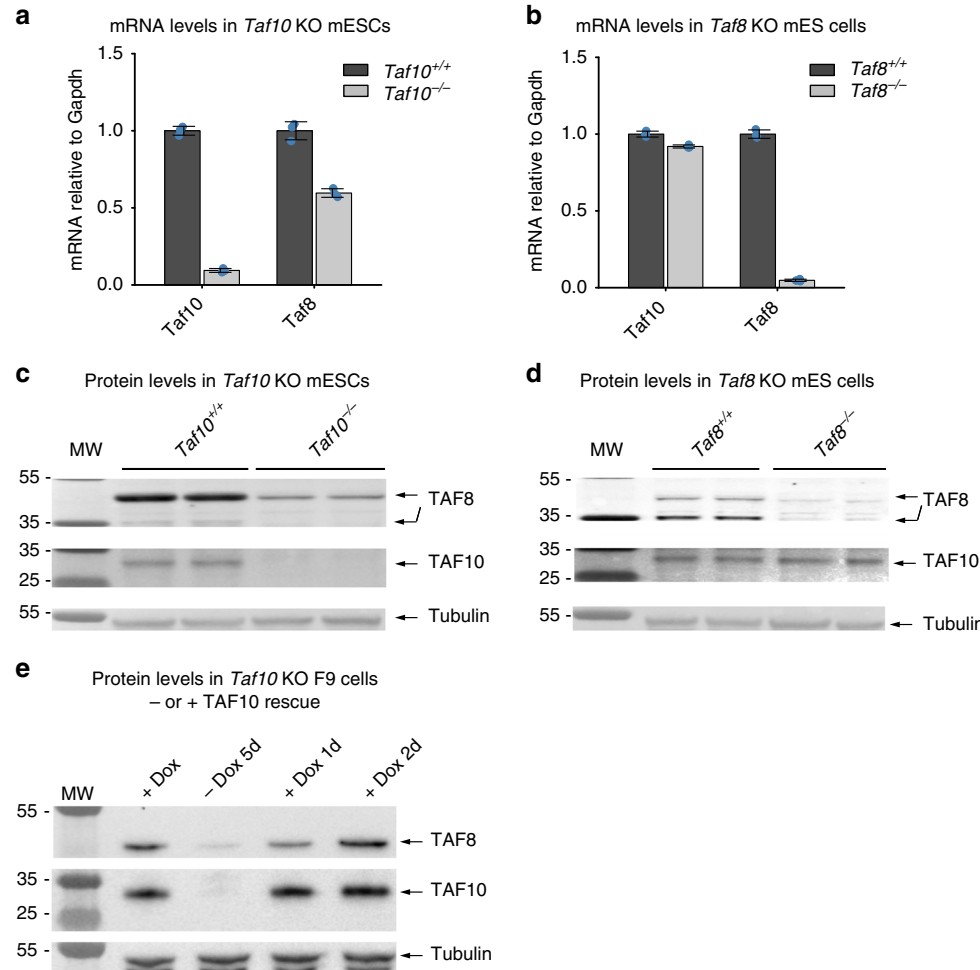

**Fig. 4** TAF10 depletion causes degradation of TAF8 in KO mESCs. **a**, **b** RT-qPCR of TAF10 (**a**) and TAF8 (**b**) depleted mESCs. **c**, **d** Western blot analyses from TAF10 (**c**) and TAF8 (**d**) depleted mESCs whole-cell extract using anti-TAF8 and anti-TAF10 antibodies. **e** Western blot analysis of whole cell extracts prepared from the mouse *Taf10*$^{-/-}$ F9 cells with or without Doxycycline (Dox) for the indicated number of days (**d**) using anti-TAF8 and anti-TAF10 antibodies. In panels (**a**, **b**), mRNA levels were normalised to *Gapdh* mRNA and relative enrichment was calculated using the ΔΔCp method and error bars represent ±SD from three technical replicates (represented by blue dots). In panels (**c**–**e**), molecular weight (MW) markers are shown in kDa and an anti-Tubulin was used as a loading control. In panels (**c**–**d**), the two protein isoforms of mTAF8 are indicated. Source data provided as a Source Data File

Together, these results further indicate that TAF10 interacts co-translationally with nascent TAF8 and when TAF10 is not present both TAF8 protein and mRNA could be prone to degradation. Thus, the nascent TAF8 HFD, in the absence of its interaction partner TAF10, may serve as a signal for both protein and mRNA degradation, while TAF10 is stable in the absence of TAF8. However, the reduction of *TAF8* mRNA in the absence of TAF10 protein due to primary transcriptional response cannot be ruled out.

**TAF10 protein co-localises with *TAF8* mRNA in the cytoplasm**. To visualise the co-localisation of TAF10 protein with *TAF8* mRNA in the cytoplasm, we set out to detect TAF10 protein and *TAF8* mRNA in the cytoplasm of fixed human HeLa cells. To this end we combined protein detection by immunofluorescence (IF) with RNA detection by single molecule inexpensive FISH (smiFISH)[23]. Co-localization of protein and mRNA was then observed by confocal microscopy and quantified. Surprisingly, we observed a large difference between the number of total (nuclear and cytoplasmic) endogenous *TAF8* and *TAF10* mRNAs, showing that there are about four times less *TAF8* mRNAs than those of *TAF10* (Supplementary Fig. 4a, b). In good agreement with our

above endogenous anti-TAF10 RIP results (Fig. 1d, e), these IF-smiFISH experiments showed an about 10% co-localization between *TAF8* mRNA and TAF10 protein in the cytoplasm of HeLa cells (Supplementary Fig. 4c). To increase the number of *TAF8* mRNA molecules in the cytoplasm of HeLa cells and to be able to carry out analyses with-wild type (wt) and mutant (mt) TAF8 proteins, we carried out IF-smiFISH detections in HeLa cells exogenously expressing TAF8 protein. The IF-smiFISH co-localization experiments in fixed HeLa cells showed significant co-localisation between TAF10 protein and *TAF8* mRNA in the cytoplasm (Fig. 5a, e; note that to observe only the cytoplasmic IF signals the nuclear signal in the green channel was removed). Importantly, the co-localization between *mtTAF8* mRNA (Fig. 3a) and TAF10 protein was lost (Fig. 5b, e). In addition, TAF8 protein detection by IF and *TAF10* mRNA by smiFISH, showed no significant co-localisation (Fig. 5c, e). Moreover, we could not detect any co-localisation between *CTNNB1* (*catenin beta-1*) mRNA and TAF10 protein (Fig. 5d, e), which further rules out any non-specific co-localisation of TAF10 protein with wt *TAF8* mRNA. Importantly, the statistical analysis of the co-localization enrichment ratio of TAF10 protein-wt *TAF8* mRNA measured in cells was significantly higher compared with all the

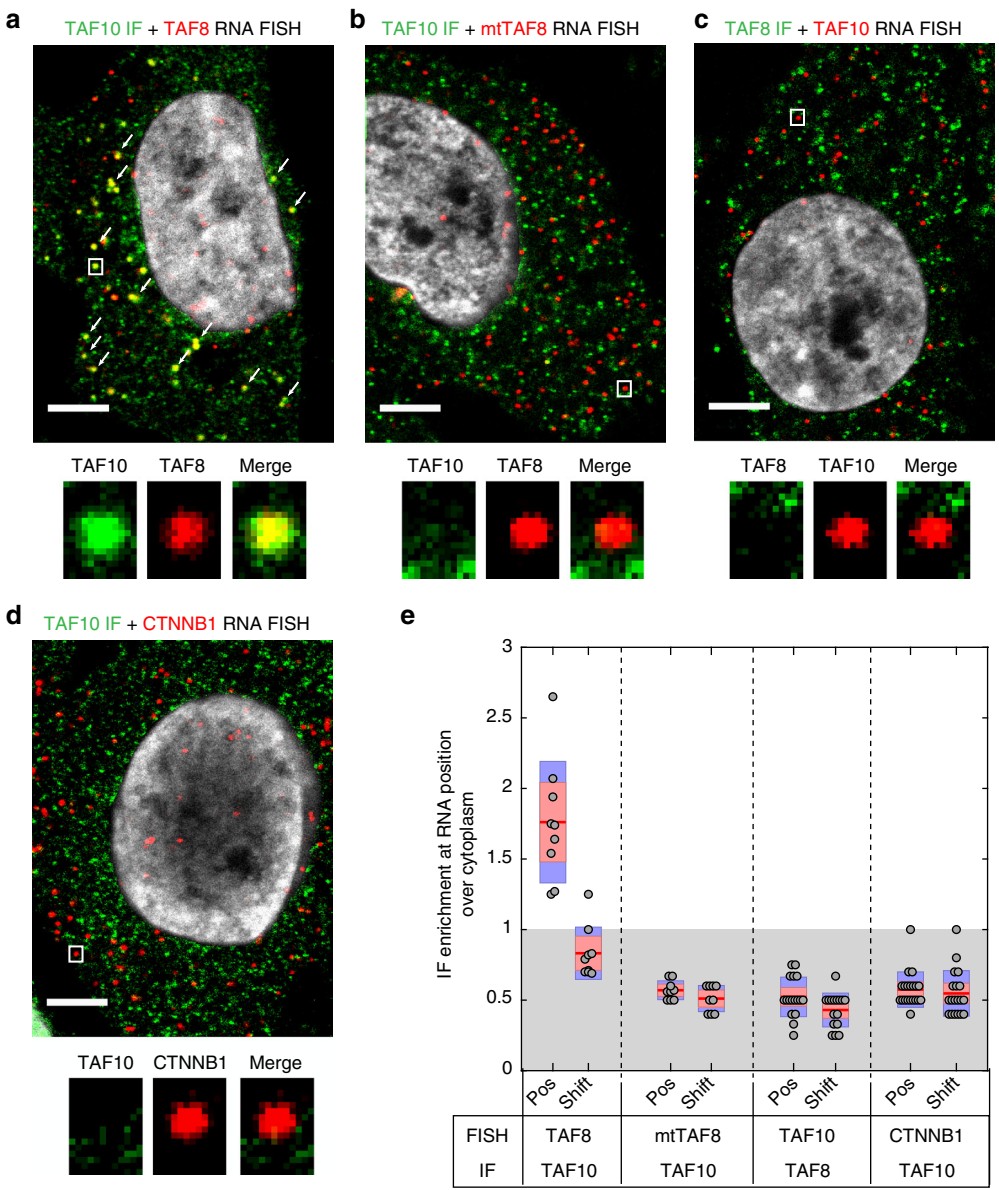

**Fig. 5** Co-localization of TAF10 protein and *TAF8* mRNA in cytoplasm. **a, b** IF-smiFISH images in HeLa cells expressing either wild-type FLAG-TAF8 or mutant (mt) FLAG-TAF8 (TAF8 S57F L65F). Labels: red, Cy3-labelled *TAF8* probes; green, Alexa-488 labelled secondary antibody for TAF10 protein; co-localizing spots are indicated with white arrows. **c, d** Representative IF-smiFISH images of endogenous *TAF10* mRNA and TAF8 protein (**c**) and *CTNNB1* mRNA and TAF10 protein (**d**) in HeLa cells. Labels: red, Cy3-tagged *TAF10* FISH probes (**c**) and *CTNNB1* probes (**d**); green, Alexa-488 labelled secondary antibody for TAF8 (**c**) and TAF10 (**d**) protein. A typical cell recorded in each case and after counterstaining the nucleus with DAPI (grey) is shown. The nuclear signal in the green channel (TAF10 or TAF8 IF) was removed by masking the nucleus and using the "clear" option in ImageJ. Zoom-in regions shown under every image are indicated with a white rectangle. Scale bar (5 μm). **e** Boxplot showing enrichment ratios of IF signal at each RNA position over mean cytoplasmic intensity under all the conditions tested. Each grey dot represents one cell. Red horizontal lines are mean values, 95% confidence interval is shown in pink, and standard deviation in blue

other conditions tested (Fig. 5e). These imaging experiments demonstrate the physical proximity of TAF10 protein to *TAF8* mRNA in the cytoplasm. Moreover, this proximity is dependent on the ability of the two proteins to interact, lending further support to the sequential assembly model.

**Position of IDs define the co-translational assembly pathway.**
To further test whether domain position guides co-translational assembly of HFD pairs in TFIID, TAF8 and TAF10 expression vectors were constructed in which the respective HFDs were

exchanged. Our nascent RIP experiments from cells co-transfected with these swapped cDNA constructs (*TAF10-HFD8* and *TAF8-HFD10*) resulted in comparable *TAF8-HFD10* mRNA and protein enrichments (Fig. 6a, b and Supplementary Fig. 5a, b); as observed with the corresponding wt constructs (Fig. 2a, b), indicating that the origin of the HFD does not influence the sequential order of co-translational assembly. This experiment also suggested that the position of the HFD (N- or C-terminal), but not its sequence, determines the co-translational pathway by which the protein partners interact. Thus, next we tested whether

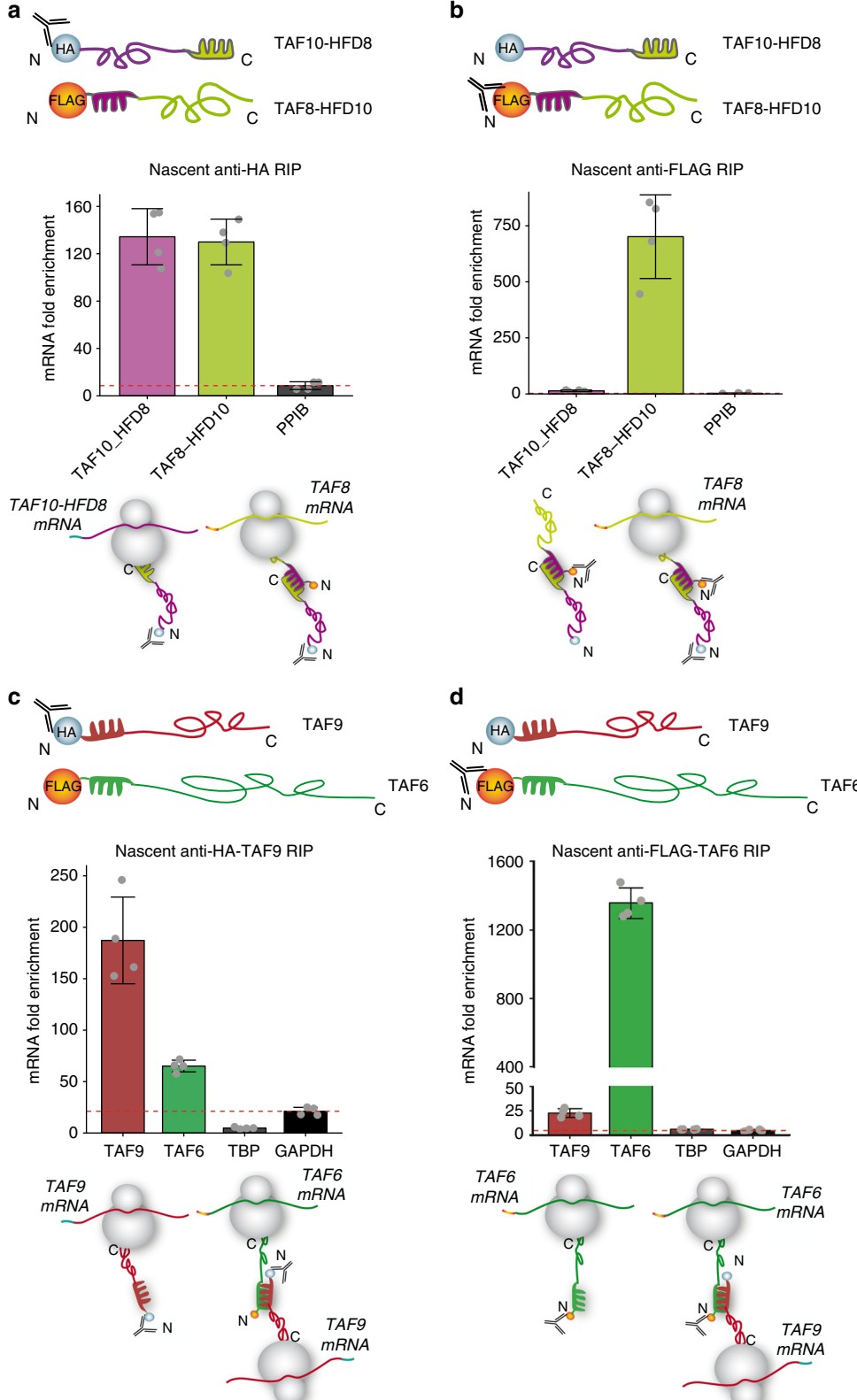

**Fig. 6** ID position determines the order of co-translational assembly. RIP-qPCR of HFD domain-swapped TAF10 and TAF8 expression constructs using anti-HA (**a**) or anti-FLAG (**b**) antibodies against the respective N-terminal tags. **c**, **d** RIP-qPCR of anti- HA-TAF9 IP (**c**) and anti-FLAG-TAF6 IP (**d**) from HeLa cell polysome extracts co-transfected with TAF9 and TAF6 expression constructs as indicated. *PPIB* (**a**, **b**) and *GAPDH* (**c**, **d**) were used as negative control mRNAs. In all the graphs error bars represent ±SD from two biological replicates and two technical replicates (represented by grey dots). Source data provided as a Source Data File

the co-translational assembly of TAF6-TAF9 HFD pair would follow the simultaneous pathway, as they interact through their N-terminal HFDs (Fig. 6c, d). Our nascent RIPs revealed that both TAF6 and TAF9 co-IP their partners' mRNA (Fig. 6c, d and Supplementary Fig. 5c, d), suggesting that they assemble through the simultaneous assembly pathway, presumably as the neo-synthesised interaction domains of both proteins are exposed early during their synthesis on the ribosomes. Such a model would further suggest that *TAF6* and *TAF9* mRNAs could be found in close vicinity in the cytoplasm. To test the simultaneous co-translational assembly of TAF6-TAF9 HFD pair we have carried two colour smiFISH co-localization experiments to detect *TAF6* or *TAF9* mRNAs in fixed HeLa cells. These experiments showed a significantly higher co-localisation of the *TAF6* and *TAF9* mRNAs in the cytoplasm than several unrelated negative control mRNAs (Supplementary Fig. 6). These results show that the simultaneous co-translational assembly of TAF6-TAF9 HFD is detectable in the cytoplasm, however, with a relatively low frequency. This can be potentially explained by the fact that TAF6 can interact with TAF9b, and TAF9 with TAF6L[13], but the corresponding *TAF6L* and *TAF9b* mRNA co-localization combinations were not tested. Moreover, we cannot rule out the possibility that the fully synthesised TAF6 or TAF9 could also find their respective nascent partners still bound to the ribosomes through the sequential assembly pathway. Thus, it seems that the position of the dimerization domain may play a critical role in defining the order of co-translational assembly pathway of the corresponding interacting factors.

**TBP and TAF1 interact also co-translationally.** In TFIID, the evolutionary conserved core domain of TBP interacts with TAF1 via N-terminal TAND region of TAF1 and this interaction modulates the DNA-binding activity of TBP within TFIID[24,25]. To investigate co-translational assembly of other non-HFD-dependent interactions, we carried out genome-wide microarray analysis of TBP-associated RNAs from HeLa cell polysome extracts using a monoclonal antibody against the N-terminus of endogenous human TBP. In addition to TBP mRNA, we detected strong enrichment of 19 coding and non-coding RNAs. Among these, we found mRNAs coding for known TBP-interacting proteins: *BRF1* coding for a factor important for Pol III transcription[26], *BTAF1* coding for a B-TFIID subunit[27], as well as *TAF1*, whose enrichment on the microarray was somewhat weaker (Fig. 7a). Nevertheless, RIP-qPCR analysis in human HeLa cells (Fig. 7b) and mouse ESCs (Supplementary Fig. 7a) confirmed the microarray data and revealed a strong enrichment of the *TAF1* mRNA. Quantification of the *TAF1* mRNA in the anti-TBP RIP normalised to the protein IP efficiency indicated that the *TAF1* mRNA enrichment was around 62%. Consistent with the need for active translation, enrichment of all specific mRNAs was lost, or greatly decreased, upon puromycin treatment (Fig. 7b).

To further investigate the specificity of TBP-TAF1 interaction, we co-transfected expression vectors coding for the full-length human TBP with a ΔTAF1 expression vector, in which sequences coding for the first 168 residues containing the TAND region were deleted. Anti-TBP RIPs from cells expressing ΔTAF1 resulted in complete loss of *TAF1* mRNA enrichment and a reduction of the co-immunoprecipitated protein (Fig. 7c, d, Supplementary Fig. 7b, c). These results are consistent with a requirement of the N-terminal TAF1 domain to recruit TBP to the nascent TAF1 polypeptide. As the protein interface is formed by the C-terminal portion of TBP and the very N-terminus of TAF1[25,28], we predicted that similarly to TAF8-TAF10 assembly, a sequential assembly is also involved in the TBP-TAF1

interaction. Indeed, nascent anti-TAF1 RIP from an engineered GFP-TAF1 HeLa cell line (Fig. 7e, f) resulted in the enrichment of *TAF1* mRNA, but not that of *TBP*, thus supporting the co-translational assembly of TBP-TAF1 by the sequential pathway.

**TREX-2 and SAGA DUB complexes assemble co-translationally.** To extend our findings beyond TFIID, we examined co-translational assembly of ENY2 subunit with its respective partners. ENY2 is subunit of the TREX-2 mRNA-export complex and the DUB module of the SAGA transcription coactivator[13]. In TREX-2, two ENY2 proteins wrap around the central portion of the large GANP helical scaffold[29]. Similarly, human ENY2 wraps around the N-terminal helix of human ATXN7L3 in the highly intertwined SAGA DUB module[30] (Fig. 8a). To test whether the co-translational model is generally applicable to multisubunit complexes, we analysed ENY2-associated mRNAs from HeLa cells stably expressing ENY2 with an N-terminal GFP-tag[31]. Interestingly, we found that an anti-GFP-ENY2 RIP co-immunoprecipitates predominantly endogenous *GANP* mRNA and protein (the partner of ENY2 in TREX-2), and also endogenous *ATXN7L3* mRNA and protein (the binding partner of ENY2 in the SAGA DUB module) (Fig. 8b, c). Together, these results demonstrate that co-translational assembly is involved in the assembly of mammalian transcription complexes of diverse architecture and function.

## Discussion

A functional protein must fold, translocate to its site of action and assemble with the right partners to carry out its function in the cell. The folding and assembly should be a well-regulated process in the cell to avoid non-specific interactions, and also because a single protein might interact with various partners depending on its interaction domain. Most eukaryotic proteins have more than one domain, which enables them to associate with their interaction partners. The building of multi-protein complexes in eukaryotes necessitates co-translational protein folding, the folding of a particular ID while still attached to translating ribosomes, to increase the efficiency of protein synthesis and prevent non-productive interactions[32]. Importantly, co-translational folding is aided by the ribosome, which stabilises specific folding intermediates of a protein[33–35]. Our results further demonstrate that the co-translational dimerization of protein interaction domains directs the assembly of mammalian nuclear multisubunit complexes. The cytoplasmic IF-smiFISH experiments indicate that the described co-translational assembly is clearly occurring in the cytoplasm of human cells and together with the mRNA enrichment calculations show that co-translational assembly is not a minor event. We also show that the position of the heterodimerization domain in a protein could guide its co-translational assembly either by sequential or simultaneous pathways. These mechanisms could play an important role in maintaining cellular health as excess orphan protein subunits can overburden protein folding and quality control machineries[36]. There is a strong correlation between the amino acid sequence of a protein, its translation rate and co-translational folding[37]. Rare codons in the mRNAs decrease the rate of translation, thereby allowing the protein to fold co-translationally[33]. Interestingly, translation pause sites are located downstream of the ID boundaries in order to regulate proper folding of multi-domain proteins[38], probably by assuring enough time for the co-translational interaction between the interacting subunits. In good agreement, our *Taf10* and *Taf8* KO mESCs, as well as F9 TAF10 ablation/re-expression experiments suggest that if the nascent ID exiting from the synthesizing ribosome cannot bind with its partner, the lack of interaction will lead to its

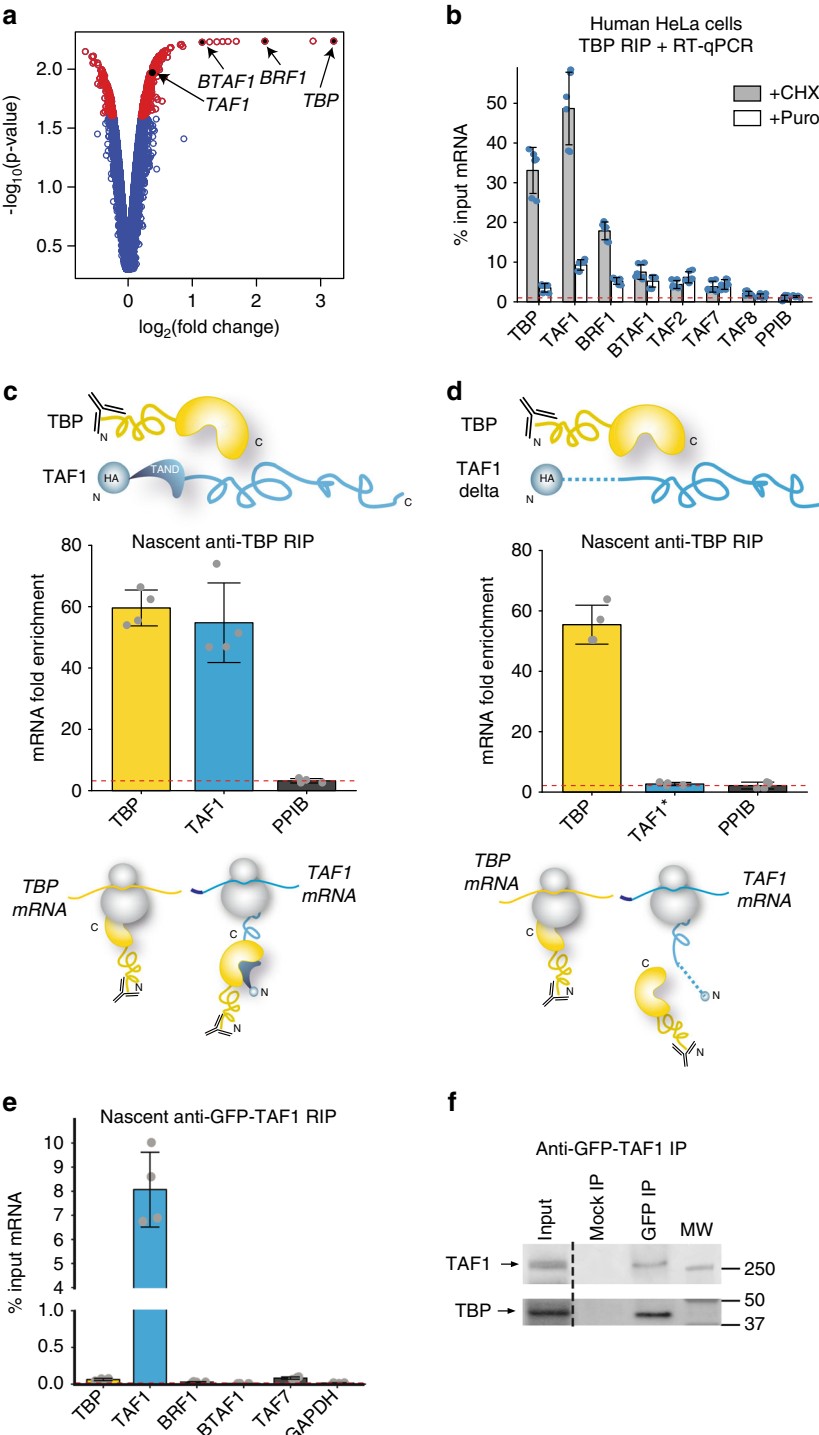

**Fig. 7** Co-translational assembly of TBP and TAF1. **a** Microarray results of RIP with an antibody targeting the N-terminus of the endogenous TBP protein. Volcano plot depicting microarray results as log2 of the fold change of IP over mock. A *p*-value cut-off ≤ 0.025 was applied and corresponding transcripts are in red. Transcripts of interest are highlighted in black. **b** RIP-qPCR validation of microarray results in HeLa polysome extracts. **c**, **d** RIP-qPCR using anti-TBP antibody in HeLa cells transfected with TBP and HA-TAF1 expression constructs (**c**) or with TBP and HA-TAF1 with N-terminal deletion of the first 168 amino acids (**d**). **e** Anti-GFP RIP-qPCR from polysomes of HeLa cells stably expressing GFP-TAF1. **f** Western blot analysis (WB) of GFP IP from polysome extracts prepared from GFP-TAF1 cell line. In (**f**) the dotted line indicates the cutting out of unnecessary lanes. All error bars are ±SD from three (**b**) or two (**c**–**e**) biological replicates and two technical replicates [represented by blue dots in (**b**) or grey dots (**c**–**e**)]. Molecular weight (MW) markers are shown in kDa. *PPIB* (**b**–**d**) and *GAPDH* (**e**) were used as unrelated control mRNAs. Source data provided as a Source Data File

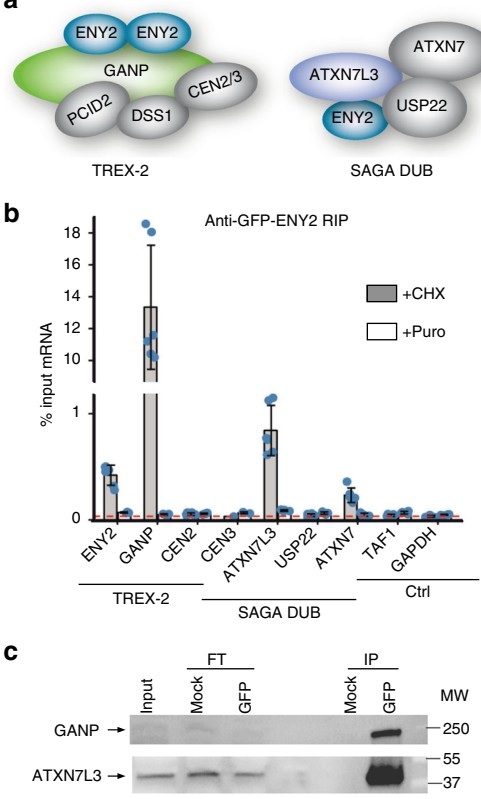

**Fig. 8** TREX-2 and SAGA DUB complexes also assemble co-translationally. **a** ENY2 is shared between TREX-2 and SAGA complex. Two protein molecules of ENY2 binds to the large GANP subunit of TREX-2 complex and ENY2 is also a part of the deubiquitination module of the SAGA complex. **b** RIP-qPCR of GFP IP from HeLa cell polysome extracts stably expressing GFP-ENY2 tagged at the N-terminus. Error bars represent ±SD from two biological replicates and two technical replicates (represented by blue dots). **c** Western blot analysis of GFP IP from polysome extract prepared from GFP-ENY2 cell line. Molecular weight (MW) markers are shown in kDa. FT flow-through. IP immunoprecipitation. *GAPDH* was used as unrelated control mRNA. Source data provided as a Source Data File

translational arrest and consequent degradation of both the nascent protein and possibly the mRNA coding it. Note that the translational pausing causing mRNA destabilization could be an attractive model, however, primary transcriptional instead of posttranscriptional response cannot be ruled out. Nevertheless, it is conceivable that when nascent IDs are translated, the ribosome may pause or slow down until the interaction partner would arrive and bind, and thus stabilise the nascent ID. However, further systematic studies need to be carried out in order to study the role of translational pausing in co-translational protein assembly.

Co-translational assembly in homomeric proteins can also cause premature assembly of protein complexes, if two interacting nascent chains are in close proximity. It has been suggested that homomeric protein IDs are enriched toward the C termini of polypeptide chains across diverse proteomes[39] and this ID localisation is essential to prevent the assembly of homomeric proteins before proper folding. In contrast, our preliminary bioinformatics analyses using a limited curated interaction database[39] suggest that in heterodimeric proteins the N-terminal interaction regions are enriched, further underlining the idea that co-translational protein assembly in heterodimeric proteins is beneficial for assembling cellular machineries.

The role of chaperones in ribosome-associated nascent protein folding is well studied. Hsp70 family of proteins (such as, e.g., yeast Ssb) protects the nascent polypeptide from misfolding and aggregation in eukaryotes[39,40]. In bacteria and yeast, the ribosome-associated chaperones have been shown to interact with the nascent polypeptide chain emerging from the ribosome aiding in its folding[8,41–43]. Moreover, recently it has been suggested that upon emergence of a complete ID, the nascent chain interacts with its partner subunit and dissociates the chaperone complex from the nascent chain[8].

Our results reveal a systemic co-translational building of complexes in mammalian cells, but a thorough proteomic approach is necessary to identify chaperones necessary for these assembly pathways. It is possible that some of the chromatin regulatory complexes assemble through other chaperone-based mechanisms in the cytoplasm or directly in the nucleus.

In summary, we show that building blocks of mammalian nuclear transcription complexes, such as TFIID, SAGA and TREX-2, are assembled during translation and the way in which assembly occurs is consistent with the current knowledge of the preliminary structural organization of the complexes. Similar results from yeast, mouse, and human cells demonstrate that co-translational assembly is a general mechanism in eukaryotes [ref. 8 and this study]. Thus, the co-translational assembly of multi-protein complexes pathways seems to be a common regulatory mechanism in all eukaryotic cells to ensure efficient solutions to avoid non-specific protein interactions, protein aggregation and probably also to control the correct stoichiometry of subunits belonging to distinct complexes. In addition, our findings will significantly advance structural biology studies, because in the future extensive screening experiments will not be required to identify a real interaction partner(s) of a given subunit in a multi-protein complex. It will be enough to make an anti-subunit RIP from polysome extracts coupled to microarray analyses (or to RT-qPCRs) and the real endogenous interacting partner(s) can be taken immediately with high confidence for structural determinations and for building the architecture of multi-protein complexes.

## Methods

**Antibodies**. Sources, catalogue numbers and concentrations of antibodies used for RIP, protein IP and western blotting are summarised in Supplementary Table 2.

**Preparation of polysome-containing extracts and RIP**. Polysome-containing extracts were prepared from adherent cells harvested at ~90% confluence by adapting a method for the isolation of ribosomes translating cell type-specific RNAs[44]. Briefly, 10 cm plates were treated with cycloheximide (100 μg/ml final) or puromycin (50 μg/ml final) and returned to the 37 °C incubator for 15 or 30 min, respectively. Subsequently, plates were placed on ice, washed twice with ice-cold PBS and scraped in 500 μl lysis buffer (20 mM HEPES KOH pH 7.5, 150 mM KCl, 10 mM MgCl₂ and 0.5% (vol/vol) NP-40), supplemented with complete EDTA-free protease inhibitor cocktail (Roche), 0.5 μM DTT, 40 U/ml RNasin (Promega), and cycloheximide or puromycin as needed. Extracts were prepared by homogenizing cells by 10 strokes of a B-type dounce and centrifugation at 17,000 × g. Clarified extracts were used to start immunoprecipitations, after saving 10% total RNA for input measurement. For TAF10 and TBP IPs, 20 μl of Protein G Dynabeads (ThermoFisher Scientific) were equilibrated by washing three times in lysis buffer, resuspended in 400 μl of lysis buffer and 2 μl of antibody, and incubated for 1 h at room temperature with end-to-end mixing. Beads were washed twice with IP500 buffer (20 mM Tris-HCl, pH 7.5, 150 mM KCl, 10% glycerol (v/v) and 0.1% NP-40 (v/v)) and three times in lysis buffer. Antibody-bound beads were thus used to perform RIP with polysome extracts overnight at 4 °C with end-over-end mixing. Mock RIP was carried out with equal amount of anti-GST antibody. The next day, beads were washed four times for 10 min at 4 °C with high salt-containing wash buffer (20 mM HEPES-KOH pH 7.5, 350 mM KCl, 10 mM MgCl₂ and 0.1% (vol/vol) NP-40) and subsequently eluted in 350 μl RA1 Lysis buffer and 7 μl 1 M DTT. RNAs were purified according to the manufacturer's instructions of the Macherey-Nagel total RNA purification kit, including the optional on-column DNase digestion step, and eluted twice in the same 60 μl of RNAse-free water. In the case of FLAG, HA, or GFP RIPs, 50 μl packed anti-FLAG M2 affinity gel (Sigma), 50 μl

packed EZview™ Red Protein A affinity gel (Sigma) or 30 µl GFP-TRAP (Chromotek) slurry were equilibrated in lysis buffer and used for RIP.

**cDNA preparation and RT-qPCR.** For cDNA synthesis, 5 µl of purified RIP-RNA and 5 µl of 1:10 diluted input RNA samples were used. cDNA was synthesised using random hexamers and SuperScript IV (ThermoFischer Scientific) according to the manufacturer's instructions. For RIP performed on transfected cells, RNA was additionally treated with Turbo DNase (Ambion) according to the manufacturer's instructions in order to ensure complete plasmid removal before cDNA synthesis. Quantitative PCR was performed with primers (listed in Supplementary Table 1) on a Roche LightCycler 480 instrument with 45 cycles. In all cases, control cDNAs prepared without reverse transcriptase (−RT) were at least over 10 Cp values of the +RT cDNAs. Enrichment relative to input RNA was calculated using the formula $100 \times 2^{[(Cp(Input) - 6.644) - Cp(IP)]}$ and expressed as "% input RNA". In the case of RIPs performed on transfected cells, enrichment values were expressed as "mRNA fold enrichment" relative to the mock IP using the formula $\Delta\Delta Cp$ [IP/mock], to account for the variability of transient transfections. "Relative mRNA fold enrichment" is expressed as mRNA fold enrichment of *TAF8* relative to mRNA fold enrichment of *TAF10* mRNA. All experiments were performed with a minimum of two biological and two technical replicates and values are represented as mean ±SD. Figures panels were prepared with taking in account all these data points using R (RStudio version 1.1.456 and R version 3.5.1).

**Microarray analysis and library preparation.** Polysome extracts and RIP from HeLa cells were performed as described above with mouse monoclonal antibodies 1H8 targeting the N-terminus of TAF10, 3G3 targeting the N terminus of TBP, and 1D10 targeting GST as a nonspecific control (see Supplementary Table 2). Protein G Sepharose beads were used (100 µl beads coupled to 14 µl antibody). After quantification and quality controls performed on Agilent's Bioanalyzer, biotinylated single strand cDNA targets were prepared, starting from 200 ng of total RNA, using the Ambion WT Expression Kit (Cat # 4411974) and the Affymetrix GeneChip® WT Terminal Labelling Kit (Cat # 900671) according to Affymetrix recommendations. Following fragmentation and end-labelling, 3 µg of cDNAs were hybridised for 16 h at 45 °C on GeneChip® Human Gene 2.0 ST arrays (Affymetrix) interrogating over 40000 RefSeq transcripts and ~11,000 LncRNAs represented by ~27 probes spread across the full-length of the transcript. The chips were washed and stained in the GeneChip® Fluidics Station 450 (Affymetrix) and scanned with the GeneChip® Scanner 3000 7 G (Affymetrix) at a resolution of 0.7 µm. Raw data (.CEL Intensity files) were extracted from the scanned images using the Affymetrix GeneChip® Command Console (AGCC) version 4.0. CEL files were further processed with Affymetrix Expression Console software version 1.3.1 to calculate probe set signal intensities using Robust Multi-array Average (RMA) algorithms with default settings (Sketch quantile normalization). Statistical analysis was performed using the FCROS package version 1.5.4[45]. Differences are considered significant for *p* value below 0.025. Volcano plots were performed using RStudio software version 3.3.2. Ribosomal RNA transcripts were filtered out. The microarray results reported in this paper are available in the Gene Expression Omnibus (GEO) under accession number GSE106299.

**Cell lines, cell culture and transfections.** HeLa cells (ATCC® CCL-2™) grown on adherent plates were obtained from the IGBMC cell culture facility and cultured in a 37 °C humidified/5% CO2 incubator. Culture media consisted of Dulbecco's modified Eagle's medium (DMEM), supplemented with 1 g/l glucose, 5% fetal calf serum (FCS), and 40 µg/ml Gentamycin. The GFP-TAF1 cell line was generated by transferring full length human TAF1 fused at its N-terminus to EGFP into HeLa Flp-In/T-REx cells following procedures described in ref. [46]. E14 mouse embryonic stem cells [mESCs, ES Parental cell line E14Tg2a.4, obtained from Mutant Mouse Resource and Research Center (MMRRC), Citation ID:RRID:MMRRC_015890-UCD] at passage 29-31 were obtained from the IGBMC cell culture facility and cultured on gelatinised plates in feeder-free conditions in KnockOut DMEM (Gibco) supplemented with the following: 20 mM L-glutamine, Pen/Strep, 100 µM non-essential amino acids, 100 µM β-mercaptoethanol, N-2 supplement, B-27 supplement, 1000 U/ml LIF (Millipore), 15% ESQ FBS (Gibco) and 2i (3 µM CHIR99021, 1 µM PD0325901, Axon MedChem). Cells were passaged approximately every 3 days. The EGFP-ENY2 HeLa cell line was generated in our laboratory by D. Umlauf[31] and maintained at 37 °C in DMEM (1 g/l glucose), 10% FCS and 40 µg/ml Gentamycin[31]. The Dox-inducible hTAF10 expression system in *Taf10*[−/−] mouse F9 embryonal carcinoma cells was generated in our laboratory by E. Scheer[22]. Cells were cultured at 37 °C with 7% CO2 in gelatinised plates in a culture media consisting of DMEM (4.5 g/l glucose), 10% FCS, 40 µg/ml Gentamycin in the presence of doxycycline (Sigma). The EGFP-ENY2 HeLa and the *Taf10*[−/−] mouse F9 embryonal carcinoma cell lines are available upon request.

Transfections were performed on ~90% confluent cells in 10 cm plates in antibiotic-free media using Lipofectamine 2000 (Thermo Fisher Scientific) and 3 µg plasmid DNA, according to the manufacturer's instructions. The medium was replaced with fresh medium containing gentamycin ~5–6 h post transfection and cells were harvested 24 h later. A descriptive summary of the plasmids used is presented in Supplementary Table 3.

**Protein IP and western blot.** Antibodies used for RIP, protein IP and western blotting are summarised in Supplementary Table 2. For protein IP, the procedure was performed essentially as for RIP. Bound proteins were eluted in 2× Laemmli buffer supplemented with 20 mM DTT and boiled for 5 min. Subsequently, samples were resolved on SDS-PAGE gels and transferred to nitrocellulose membranes using either wet transfer or BioRad's Trans-Turbo Blot semi-dry transfer method. Secondary antibodies (goat anti-mouse or rabbit anti-mouse) coupled to HRP (Jackson ImmunoResearch Laboratories) were used at 1:10,000 dilution. Signal was revealed using chemiluminescence (Pierce) and detected on the ChemiDoc imaging system (BioRad). For immunoprecipitation using whole cell extracts, 10 confluent 10 cm plates were scraped in PBS containing protease inhibitor (Roche) and resuspended in ~1 packed cell volume lysis buffer (20 mM Tris-HCl, pH 7.5, 400 mM KCl, 2 mM DTT, 20% glycerol) supplemented with protease inhibitor and 0.5 mM final concentration of DTT. Extracts were prepared by four cycles of freezing on liquid nitrogen followed by thawing on ice. The concentration of the clarified extract was measured by Bradford assay and the extract was diluted ~1:3 using lysis buffer without salt to achieve a final concentration of ~150 mM KCl. One-milligram extract was added to mock- and antibody-bound beads each and IPs were performed as described above. Proteins were eluted twice for 5 min at room temperature in 50 µl 0.1 M Glycine, pH 2.8 and neutralised with 3.5 µl 1.8 M Tris-HCl, pH 8.8. Ten percent of the pooled eluates were resolved on gels.

**Plasmids.** The eukaryotic expression plasmid pXJ41 used for all the constructs has been previously described[47]. pXJ41-TAF10-Nter-2HA has been previously described[48]. To generate N- and C-terminally Flag-tagged TAF8, the human TAF8 cDNA was PCR amplified from pACEMam1-CFP-TAF8 (kind gift from Imre Berger, University of Bristol, UK) using primers cotaining EcoR I and Bgl II restriction sites and tags incorporated at the N- or C-terminus, respectively, and digestion by appropriate restriction enzymes. Similarly, C-terminal HA tagged TAF10 was subcloned from pXJ41-TAF10-Nter-2HA by PCR amplification and digestion via restriction enzymes Xho I and Kpn I. The TAF8 mutations, TAF8-HFD and TAF8-HFD-60 amino acids were generated by site-directed mutagenesis using PfuUltra High-Fidelity DNA polymerase (Agilent Technologies), according to the manufacturer's instructions. The histone fold domain swapped TAF10 and TAF8 constructs were generated with several rounds of PCR amplification, using the already-mentioned N-terminal tagged TAF10 and TAF8 constructs as a template with specific primers and cloned into the vector via restriction enzymes EcoR I and Bgl II. pXJ41-hTBP has been previously described[49]. The HA-TAF1 cDNA[50] was inserted in pXJ41. TAF1 N-terminal deletion was carried out by site-directed mutagenesis using PfuUltra High-Fidelity DNA polymerase (Agilent Technologies), according to the manufacturer's instructions. HA tagged TAF9 was subcloned from pSG5-TAF9[51] by PCR amplification and digestion by restriction enzymes EcoR I and Bgl II. FLAG-tagged TAF6 was also subcloned in a similar manner from pXJ41-TAF6[52] via restriction enzymes Xho I and Kpn I. All plasmids have been verified by sequencing. Details on the cloning strategies are available upon request. Plasmids are described in Supplementary Table 3.

**Mouse *Taf8* and *Taf10* KO ESC lines.** The *Rosa26*[Cre-ERT2/+]; *Taf8*[flox/flox] mouse embryonic stem cells (mESCs) were generated previously by F. El Saafin[21]. Briefly, mice carrying the *Taf8*[lox] allele were bred to mice carrying the *Rosa26*[Cre-ERT2] allele to produce *Rosa26*[Cre-ERT2/+]; *Taf8*[flox/flox] E3.5 blastocysts and to isolate *Rosa26*[Cre-ERT2/+]; *Taf8*[flox/flox] mouse embryonic stem cells (mESCs)[21]. The *Rosa26*[Cre-ERT2/R]; *Taf10*[flox/flox] mESCs were generated previously by P. Bardot[20]. Briefly, the ESCs were derived from *Rosa26*[Cre-ERT2/R];*Taf10*[lox/lox] E3.5 blastocysts[20]. mESCs were cultured in DMEM (4.5 g/l glucose) with 2 mM Glutamax-I, 15% ESQ FBS (Gibco), penicillin, streptomycine, 0.1 mM non-essential amino acids, 0.1% ß-mercaptoethanol, 1500 U/mL LIF and two inhibitors (2i; 3 µM CHIR99021 and 1 µM PD0325901, Axon MedChem) on gelatin-coated plates. To induce deletion of *Taf8*, mESCs were treated with 0.5 µM 4-OH tamoxifen (Sigma) for 5–6 days, and to induce deletion of *Taf10*, *Rosa26*[Cre-ERT2/R];*Taf10*[lox/lox] mESCs were treated for 4 days with 0.1 µM 4-OH tamoxifen (Sigma). The above-described mESCs have already been described[20,21] and were derived according to animal welfare regulations and guidelines of the French Ministry of Agriculture and French Ministry of Higher Education and Research, and the Australian Animal Welfare Committee, respectively.

**smiFISH.** smiFISH primary probes were designed with the R script Oligostan as previously described[23]. Primary probes and secondary probes (Cy3 or digoxigenin conjugated FLAPs) were synthesised and purchased from Integrated DNA Technologies (IDT). Primary probes were ordered at a final concentration of 100 µM, wet and frozen in Tris-EDTA pH 8.0 (TE) buffer. Probe sequences are available in Supplementary Table 4. An equimolar mixture of all the primary probes for a particular RNA was prepared with a final concentration 0.833 µM of individual probes. The secondary probes are resuspended in TE buffer at a final concentration of 100 µM. A total of 10 µl of FLAP hybridization reaction was prepared with 2 µl (for single colour smiFISH) or 4 µl (for dual colour smiFISH) of diluted (0.833 µM) primary probe set, 1 µl of secondary probe, 1 µl of 10X NEB3 and 6 µl of water. The reaction mix was then incubated in a cycler under the following conditions: 85 °C, 3 min, 65 °C, 3 min, 25 °C, 5 min. Two microliters of these FLAP hybridised probes

are necessary for each smiFISH reaction. The volume of the reactions were scaled up according to the number of smiFISH reactions carried out.

smiFISH was carried out as follows as per published protocol[23]. HeLa cells were treated with 100 µg/ml cycloheximide (Merck) for 15 min at 37 °C, fixed with 4% paraformaldehyde (Electron Microscopy Sciences) for 20 min at room temperature (RT) followed by overnight incubation with 70% ethanol at 4 °C. Following overnight incubation, cells were rinsed with 1× PBS twice and incubated with Solution A (freshly prepared 15% formamide in 1× SSC buffer) for 15 min at RT. During incubation, 50 µl Mix 1 (5 µl of 20× SSC, 1.7 µl of 20 µg/µl E. coli tRNA, 15 µl of 100% formamide, 2 or 4 µl of FLAP hybridised probes, required amount of water) and 50 µl Mix 2 (1 µl of 20 mg/ml RNAse-free BSA, 1 µl of 200 mM VRC, 27 µl of 40% dextran sulfate, 21 µl of water) was prepared. Mix 1 was added to Mix 2 after proper vortexing. The total 100 µl of Mix1 + Mix2 is sufficient for two coverslips. Each coverslip was then incubated on a spot of 50 µl of the Mix in a 15 cm Petri dish with a proper hydration chamber (3.5 cm Petri dish containing 2 ml of 15% formamide/1× SSC solution) overnight at 37 °C. Following overnight incubation, coverslips were washed twice with Solution A at 37 °C for 30 min each and with 1× PBS twice for 10 min each. Coverslips with only Cy3 conjugated secondary probes are mounted with 5 µl of Vectashield containing DAPI at this step. For DIG-labelled secondary probes, cells were further permeabilised with 0.1% Triton-X100 for 10 min at RT and incubated with 0.25 µg/ml anti-digoxigenin-fluorescein Fab fragments (diluted in 1× PBS) (Roche) for 2 h at RT. Following antibody incubation, cells were mounted as before.

**IF-smiFISH**. To visualise proteins and mRNA together, we first performed immunofluorescence (IF) followed by smiFISH. Briefly, cells were treated with 100 µg/ml cycloheximide (Merck) for 15 min at 37 °C, fixed with 4% paraformaldehyde (Electron Microscopy Sciences) for 10 min at room temperature (RT), blocked and permeabilised with blocking buffer (10% BSA, 10% Triton-X-100, 200 mM VRC, 2X PBS) for 1 h at 40 °C, incubated for 2 h at RT with either anti-TAF8 (mouse monoclonal antibody (mAb) 1FR-1B6[53]; diluted 1:1000) or anti-TAF10 (mAb 6TA-2B11[53]; diluted 1:1000) antibody mix followed by incubation (RT, 1 h) with secondary antibody mix Alexa-488-labelled goat anti-mouse mAb (Life Technologies, catalogue number A-11001, diluted 1:3000). Following immunofluorescence described above, cells were fixed with 4% paraformaldehyde (Sigma) for 10 min at RT. Cells were washed with 1× PBS and incubated with wash buffer [10% Formamide (Sigma) in 2× SSC] for 10 min at RT. smiFISH was carried out as described above and see ref. [23]. Cells were mounted using Vectashield mounting medium with DAPI (Vector laboratories Inc.).

**Imaging and image processing**. Confocal imaging of smiFISH and IF-smiFISH samples was performed on an SP8UV microscope (Leica) equipped with a 633-nm HeNe laser, a 561-nm DPSS laser, a 488-nm argon laser and a 405-nm laser diode. A ×63 oil immersion objective (NA 1.4) was used and images were taken by using the hybrid detector photon-counting mode. The laser power for all acquisitions and laser lines was set to 10%. All images acquired have a bit depth of 8 bit and a pixel resolution of 70 nm. The z-stacks were taken with a z-spacing of 300 nm for a total of 4–6 µm. Image processing was performed using the Fiji/Image J software. All images were processed the same way. In detail, the channels of the different images were split and grey values were adjusted to better visualise the spots in the cytoplasm. The nuclear signal in the green channel (TAF10 or TAF8 IF) was removed by masking the nucleus and using the "clear" option. Finally, the processed channels were merged again. For IF-smiFISH, one cell of an image was cropped and one representing z-slice per cell was chosen. For smiFISH, maximum intensity Z-projections of individual images were made and one cell per resulting image was cropped as the representative image. In addition, one single IF or smiFISH spot from the corresponding cells was cropped as well.

**Image analysis of IF-smiFISH data**. To measure the degree of spatial overlap of smiFISH (mRNA) and IF (protein) signal, an enrichment ratio was calculated as described below. Such quantification was chosen in order to take into account the variability of IF signal between cells, making single object detection in this channel difficult. Cells and nuclei were outlined manually in 2D based on the GFP and DAPI image, respectively. Subsequent analyses were restricted to the cytoplasm. mRNAs were detected in 3D with FISH-quant[23]. Identical detection settings were used when different experimental conditions were compared with the same gene. Each cell was post-processed separately. First, the median pixel intensity in the IF image at the identified RNA positions was calculated. Second, a normalization factor was estimated as the median IF intensity of the outlined cytoplasm within the z-range of the detected mRNAs. The enrichment ratio of the cell was then calculated as the ratio of the median IF intensity at the RNA positions divided by the mean cytoplasmic intensity. Boxplots of enrichment ratios were generated with the Matlab function notBoxPlot. Each dot corresponds to the estimation of one cell. Horizontal lines are mean values, 95% confidence interval is shown in red, and standard deviation in blue. Statistical comparison between different experimental conditions was performed with two-sample Kolmogorov–Smirnov test (Matlab function kstest2). The Matlab script is available upon request.

**Image analysis of smiFISH co-localization data**. Segmentation of nuclei and cells was performed with the DAPI and smiFISH channel channels, respectively. 2D images were obtained with a previously described projection approach based on local and global focus measurements[23]. Segmentation was implemented with the open-source software CellProfiler[54] using a standard workflow: Otsu and watershed separation for nuclei in the DAPI channel. Each nucleus then serves as a seed for a watershed segmentation to obtain the cells in the smiFISH channel. Individual RNA molecules were localised with FISH-quant in 3D and can be treated as point clouds[55]. Co-localization analysis between detected RNAs in two colours was solved as a linear assignment problem (LAP) with the Hungarian algorithm (Matlab function hungarianlinker and munkres from Matlab FileExchange). In short, this algorithm finds the best possible global assignment between these two points-clouds such that for each point in the first colour the closest point in the second channel is found. We implemented a user interface for this analysis tool (FQ_DualColor), which is distributed together with a dedicated user manual with FISH-quant: https://bitbucket.org/muellerflorian/fish_quant

**Reporting summary**. Further information on research design is available in the Nature Research Reporting Summary linked to this article.

## Data availability
The microarray data corresponding to Figs. 1a and 7a are available in the Gene Expression Omnibus (GEO) under accession number GSE106299. The source data corresponding to Figs. 1d–e, 2a–d, 3a–b, 4a–d, 6a–d, 7b–f, 8a–c and Supplementary Figs. 1, 2a–d, 3a–b, 5a–d, 7a–c are provided as a Source Data file. A reporting summary for this Article is available as a Supplementary Information file. Raw image files (~800), their corresponding analyses, and all other data supporting the findings of the study are available from the corresponding author upon request.

## Code availability
The Matlab script (Kamenova_NatComm__rna_protein_coloc.m) concerning the RNA co-localization and IF-smiFISH analyses is available on the FISH-quant repository [https://bitbucket.org/muellerflorian/fish_quant]. The custom R scripts for dot plot overlaid bar charts are available upon request.

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

## Acknowledgements

We would like to thank all members of the Tora lab for thoughtful discussions and suggestions throughout the course of the work, E. Scheer for help with F9 cell experiment, and G. Caliskan for help with the ATAC experiments. In addition, we are grateful to E. Bertrand and his group for smiRNA FISH training, T. Sexton for advice and carefully reading the manuscript, V. Alunni for microarray sample preparation, S. Bour for illustrations, K. Gupta and I. Berger for suggesting TAF deletions and mutations, D. Van Essen for providing constructs, P. Jane Palli and G. Travé for help with protein domain analyses, the IGBMC cell culture facility for cells and media. PM was supported by a fellowship (DOC20180507393) from Fondation ARC pour la Recherche sur le Cancer. This work was supported by funds from CNRS, INSERM, and Strasbourg University. This study was also supported by the European Research Council (ERC) Advanced grant (ERC-2013-340551, Birtoaction, to L.T.) and grant ANR-10-LABX-0030-INRT (to L.T.), a French State fund managed by the Agence Nationale de la Recherche under the frame programme Investissements d'Avenir ANR-10-IDEX-0002-02 (to IGBMC) and a grant from the Collaborative Center for X-linked Dystonia and Parkinsonism (to H.T.M.T.).

## Author contributions

I.K., P.M. and L.T. designed the study; I.K. and P.M. performed all the molecular lab work (their names appear in alphabetical order in author list), J.M.G. performed all the cloning experiments. P.M. and S. Conic carried out the imaging experiments, which were analysed by F.M., S. Caponi created a stable cell line, F.E.S., P.B. and S.D.V. carried out the mouse ESC knock-out experiments; I.K. and P.M. analysed data, D.D. analysed the microarray data. All authors contributed to the text and figure panels. I.K., P.M., H.T.M.T., S.D.V. and L.T. wrote the manuscript. All authors gave final approval for publication.

## Additional information

**Competing interests:** The authors declare no competing interests.

