## [Peer Review File · Nature Communications]

Reviewers' comments:

Reviewer #1 (Remarks to the Author):

“Co-translation drives the assembly of mammalian nuclear multisubunit complexes”

Ivanka Kamenova, Pooja Mukherjee, Sascha Conic, Florian Mueller, Farrah El-Saafin, Paul Bardot, Jean-Marie Garnier, Doulaye Dembele, Simona Capponi, H.T. Marc Timmers, Stéphane D. Vincent and László TORA

This manuscript describes the results of a series of studies designed to gain insights into the mechanism of assembly of multisubunit transcription factors in mammalian cells. Specifically, do known interacting subunits of TFIID, SAGA and/or TREX complexes interact cotranslationally. Experiments were performed that analyzed the assembly of a subset of subunits from these multisubunit transcription proteins. These three holofactors exhibit complex, yet unique, subunit stoichiometries and structures. An important unanswered question in the field of structure-function studies of such complexes is how the observed unique subunit stoichiometries and corresponding structures are accurately and efficiently assembled within living cells. Here the authors have shown that for the subunits of TFIID, SAGA and TREX complexes known interacting subunits complex cotranslationally via either simultaneous or sequential interactions of the relevant subunits.

The experimental data presented is clear, compelling and well controlled. Thus, the results and conclusions are readily accepted and this study materially extends our understanding of the essential process of multisubunit transcription complex assembly and function. Indeed, given the robustness of the data obtained through the methods outlined in this report, the authors argue that one could use these methods to identify novel protein-protein interaction domains between non-histone fold protein subunits of these complexes (i.e. TFIID, SAGA, TREX) in order to facilitate expression, purification and structural analyses of modules of these and other important multisubunit transcription factors.

Considerations/Questions for Authors

1. Given the suggestions by the authors regarding the utility of their methods, it would be interesting to apply their methodologies to identify candidate proteins/domains of proteins besides the ones analyzed herein (most of which had previously been characterized with regard to ID motifs). Such

data would broaden the appeal/interest to other systems. Might the authors consider such an endeavor?

2. I am a bit confused by the results of the presented cell imaging studies where the authors use Immunofluorescence and RNA FISH to colocalize Taf10 protein and Taf8 mRNAs. Perhaps the authors could discuss why they do not observe significant IF signals over the nucleus, where one would presume the majority of TFIID and SAGA reside within the cell.

Reviewer #2 (Remarks to the Author):

Kamenova et. al perform an elegant study showing co-translational complex formation of 3 key transcriptional complexes, TFIID, TREX-2, and SAGA. First, using immunoprecipitation of TAF8 and TAF10 followed by microarray analysis, the authors show a specific enrichment of TAF8 mRNA when using TAF10 to pull down polysomes. They go on to perform the same experiment, but instead use qRT-PCR to quantify the relative enrichment of mRNAs that co-purify with the IP. They observe that the N-terminus of TAF10 pulls down both TAF8 and TAF10 mRNA. However, they were unable to use TAF8 to pull down TAF10 mRNA, leading the authors to believe that the co-translational TAF8-TAF10 complex formation may follow a sequential mechanism. These results are replicated when purified with an affinity tag on the C-terminus of these proteins. The authors go on to show that the previously mentioned co-IP results are dependent on the interaction between the HFD region of TAF8. When pulling down TAF10, a mutant TAF8-HFD domain ablates the pull down of TAF8 mRNA. Additionally, they show the TAF8-HFD domain must be clear of the ribosome exit channel in order to be recognized by TAF10. In conditional knockout mESC, the authors go on to show that TAF8 is destabilized in the absence of TAF10. Then, using smFISH-IF they show TAF10 protein colocalizes with TAF8 mRNA in cells. Lastly, they show that HFD position, and not sequence, dictate co-translational assembly. To show that a simultaneous pathway is possible, they perform RIP-IP on TAF6 and TAF9, showing that both IPs can pull down the other's mRNA. To further generalize this phenomenon, the authors interrogate ENY2 which is part of both the TREX-2 complex and the SAGA transcriptional co-activator. ENY2 co-immunoprecipitates the mRNA of its binding partners, GANP and ATXN7L3, components of TREX-2 and SAGA respectively, suggesting again there is a co-translational mechanism of assembly. Taken together, this work represents a clear advance forward in the field and rethinks how we view the translation of large multi-subunit complexes. The study is well conceived and performed with proper controls.

I have a few points need to be addressed

(1) To convincingly demonstrate that TAF6 and TAF9 are assembled through simultaneous pathway, it is necessary to demonstrate that their mRNAs are close to each other. This should be done through a simple two-color FISH against TAF6 and TAF9 respectively. Without this, it is hard to draw conclusive statements about the pathway.

(2) In the IF-smFISH experiment, why do they need to do it with transfected TAF8. The authors state that the number of TAF8 mRNA is low. But 30 mRNAs per cell are more than enough. In the analysis, the authors used intensity values, not colocalization percentage. The authors observed many big green punctas (Fig. 5a) that colocalize with transfected TAF8 mRNA. But I imagine that TAF10 would also colocalize with its own mRNAs. A similar IF-smFISH with TAF10 protein and mRNA will clarify that. Furthermore, if TAF10 is synthesized first, it will be imported into nucleus if it has nuclear localization signal (NLS). If the NLS is on TAF8, there will be many TAF10 in the cytoplasm to form diffusive signal, and it would be difficult to observe TAF10 puncta.

Minor points:

(1) In figure 3a, why is the fold enrichment of wtTAF8 mRNA only a modest $\sim 1.2x$? Is this the same experiment as Figure 2a which showed an over 50-fold enrichment? Additionally, in 3a the TAF8 mRNA is lower than that of TAF10 mRNA, which is not consistent with the wild-type result from 2a.

Reviewer #3 (Remarks to the Author):

The manuscript by Kamenova et al. contributes to exciting new developments in the field of protein biosynthesis and quality control: the cotranslational assembly of multimeric protein complexes. To study this, the authors choose model transcription complexes and utilize a combination of immunoprecipitations, microarray and sequencing, and cell biology to demonstrate that certain subunits specifically assemble cotranslationally in a defined order. Overall, the manuscript is well-presented and timely, the experiments are thoughtfully designed and executed, and the results are clear. My opinion is that the manuscript is generally well-suited for publication for Nature Communications. I only have one major comment:

To fully support the authors' claims that TAF10 knockout induces TAF8 protein degradation (Fig. 4), a simple proteasomal inhibition experiment (e.g. with MG132) should be performed to see if TAF8 protein levels are stabilized. Otherwise, the decreased protein levels could be primarily attributed to the decreased mRNA levels. Similarly, since the authors have expression constructs of TAF10, it

would be nice to see a rescue re-expression experiment in the TAF10 knockout cells to demonstrate that the regulation of TAF8 levels is acute (e.g. dependent on cotranslational assembly).

Minor comments:

1. The title does not make sense to me. "Co-translation drives the assembly..." implicates that two subunits are translated at the same time, but the authors do not directly demonstrate this. Instead, one completely synthesized subunit can interact with another that is being translated. Changing the title to "Co-translational assembly of mammalian nuclear multisubunit complexes", or something similar, seems more accurate.
2. In the abstract, the authors cite "protein translation". I suggest changing this to "protein synthesis" (mRNAs are translated, proteins are not).

Reviewers' comments:

Reviewer #1:

The experimental data presented is clear, compelling and well controlled. Thus, the results and conclusions are readily accepted and this study materially extends our understanding of the essential process of multisubunit transcription complex assembly and function. Indeed, given the robustness of the data obtained through the methods outlined in this report, the authors argue that one could use these methods to identify novel protein-protein interaction domains between non-histone fold protein subunits of these complexes (i.e. TFIID, SAGA, TREX) in order to facilitate expression, purification and structural analyses of modules of these and other important multisubunit transcription factors.

We were happy to learn that the Reviewer thought that our manuscript was “clear, compelling and well controlled” and that it “extends our understanding of the essential process of multisubunit transcription complex assembly and function”.

1. Given the suggestions by the authors regarding the utility of their methods, it would be interesting to apply their methodologies to identify candidate proteins/domains of proteins besides the ones analyzed herein (most of which had previously been characterized with regard to ID motifs). Such data would broaden the appeal/interest to other systems. Might the authors consider such an endeavor?

In our manuscript, we have investigated several HF-HF IDs, the TAF1(TAND)-TBP IDs and the ENY2-GANP and the ENY2-ATXN7L3 IDs. We agree with the reviewer that the identification of novel previously uncharacterised protein-protein interactions using our methodology would definitely broaden the appeal to other systems. But we think this would extend our findings beyond the scope of this manuscript. Nevertheless, such a systemic approach would be the basis of our next manuscript.

2. I am a bit confused by the results of the presented cell imaging studies where the authors use Immunofluorescence and RNA FISH to colocalize Taf10 protein and Taf8 mRNAs. Perhaps the authors could discuss why they do not observe significant IF signals over the nucleus, where one would presume the majority of TFIID and SAGA reside within the cell.

We apologize for the confusion in the imaging studies that may not have been well explained. We completely agree with the reviewer that the majority of TFIID and SAGA complex subunits localise to the nucleus of the cell. Since we are interested to study the co-translational assembly of the subunits of the mentioned complexes we investigated the co-localization events only in cytoplasm of the cell. In consequence, we removed the very high nuclear signal in the green channel (for both TAF10 and TAF8 IFs) by masking the nuclei and using the “clear” option in ImageJ and highlighted the DAPI signal in grey. These imaging experiments are now better explained in the manuscript (see page 8 lines 8-9, the legend of Figure 5 panels a-d, and the corresponding Methods section)

Reviewer #2

Taken together, this work represents a clear advance forward in the field and rethinks how we view the translation of large multi-subunit complexes. The study is well conceived and performed with proper controls.

We were happy to learn that the Reviewer thought that our study represents a clear advance in the field and that it is well conceived with proper controls.

(1) To convincingly demonstrate that TAF6 and TAF9 are assembled through simultaneous pathway, it is necessary to demonstrate that their mRNAs are close to each other. This should be done through a simple two-color FISH against TAF6 and TAF9 respectively. Without this, it is hard to draw conclusive statements about the pathway.

As required we have now carried out two-color smiFISH against *TAF6* and *TAF9* respectively. We do see co-localisation between the two mRNAs in the cytoplasm that is significantly higher than the two negative controls we performed. This new experiment is now described in the manuscript (see page 9 lines 6-11 and presented in Supplementary Figure 6).

Nevertheless, the colocalization is not very high. This may be explained by the fact that TAF9 interacts not only with TAF6 (in TFIID) but also with its paralogue TAF6L (in SAGA). Similarly, TAF6 interacts not only with TAF9 but also with its paralogue TAF9b (both in TFIID and SAGA). As the probes that we used for two-color smiFISH were specific to only TAF6 and TAF9, we believe that the co-localisation we observe in our experiment does not represent all TAF6/TAF6L- TAF9/TAF9b mRNA co-translational events in the cell. This point is now also discussed in the manuscript (page 9 lines 11-19).

(2) In the IF-smFISH experiment, why do they need to do it with transfected TAF8. The authors state that the number of TAF8 mRNA is low. But 30 mRNAs per cell are more than enough.

We apologize for the potential misunderstanding. In Supplementary Figure 4a we show a maximum intensity Z-projection of the whole cell including the nucleus (surrounded by a blue dotted line). When we counted the endogenous *TAF8* mRNAs in the cytoplasm we found about 20 molecules (38 for the whole cell), while for *TAF10* mRNA we find about 125 molecules in the cytoplasm. To image the cells, we take about 20 Z stacks. This also means that there will be Z stacks lacking any *TAF8* signal, while others may have several endogenous *TAF8* mRNAs, which colocalize with TAF10 protein. Such a Z-stack with endogenous *TAF8* mRNA colocalizing with endogenous TAF10 protein is shown in Supplementary Fig. 4c. However, note also that TAF10 does not only interact with TAF8, but also with TAF3 and SUPT7L (see Soutoglou et al. 2005, MCB). Thus, to increase the likelihood of encountering the situation of TAF8-10 assembly in each Z-stack and to make the number of TAF8 and TAF10 mRNAs comparable in the cytoplasm we transfected a TAF8 expression plasmid. Moreover, as we wanted to test the specificity of the co-translational assembly we needed also a wild type control for those experiments, where we would express also the HF mutated TAF8 expression vector.

In the analysis, the authors used intensity values, not colocalization percentage.

For the analysis, the smiFISH signal is comparable from cell to cell, but the IF not so much. So it is impossible to set one detection threshold for all images. To circumvent this problem, we developed a strategy that does not rely on "object" detection for the IF but rather it is intensity based. Thus, we first detected the mRNAs and then looked at the IF signal enrichment underneath these mRNAs

(compared to the rest of the cell). The analysis and its strategy is now better explained in the revised Methods section entitled “Image analysis of IF-smFISH data” (see page 20).

The authors observed many big green punctas (Fig. 5a) that colocalize with transfected TAF8 mRNA. But I imagine that TAF10 would also colocalize with its own mRNAs. A similar IF-smFISH with TAF10 protein and mRNA will clarify that. Furthermore, if TAF10 is synthesized first, it will be imported into nucleus if it has nuclear localization signal (NLS). If the NLS is on TAF8, there will be many TAF10 in the cytoplasm to form diffusive signal, and it would be difficult to observe TAF10 puncta.

We agree that a certain number of nascent endogenous TAF10 protein molecules should co-localise with their encoding TAF10 mRNAs in the cytoplasm during TAF10 protein translation. But in order to visualise this possibly very transient event, we would need a good IF grade antibody, which would bind to the N-terminal region of endogenous TAF10. The TAF10 antibody that we used for IF in Fig 5 binds to the middle region of TAF10 protein and hence it's not useful to detect the N-terminal end of the nascent TAF10 protein. However, to test the Reviewer's suggestion, we carried out IF (with an anti-HA antibody) in HeLa cells transfected with a plasmid expressing N-terminally HA tagged TAF10, but the IF signal was too high (after transfection) to perform any meaningful co-localisation studies. The Reviewer is right, TAF10 lacks an NLS (see our earlier publication by Soutoglou et al 2003, MCB) and is imported to the nucleus by TAF8, TAF3 (both TFIID subunits) or SUPT7L (a SAGA subunit). All these observations and technical issues together made the study of the colocalization of the nascent TAF10 protein and mRNA non-conclusive.

Minor points:

(1) In figure 3a, why is the fold enrichment of wtTAF8 mRNA only a modest $\sim 1.2x$? Is this the same experiment as Figure 2a which showed an over 50-fold enrichment? Additionally, in 3a the TAF8 mRNA is lower than that of TAF10 mRNA, which is not consistent with the wild-type result from 2a.

We apologize for the confusion. In Figure 2a the values are expressed as “mRNA fold enrichment” and in Figure 3a and b the results are represented as “relative mRNA fold enrichment”. These two different calculations are now better explained in the legends of Figure 2 and 3, as well as in the revised “cDNA preparation and RT-qPCR” section (see page 15). In Figure 3a and 3b, the key questions are: how do the TAF8 mutants and truncations behave compared to the wt TAF8 protein. We think that the “relative mRNA fold enrichment” representation makes it easier in the RIPs to compare the level of wtTAF8 to its mutant or truncated mRNAs under the experimental conditions, where wildtype TAF10 expression plasmid is transfected with either wild type TAF8 or mutant/truncated TAF8 expression plasmids (as depicted on the top of Fig 3a and b).

Reviewer #3

Overall, the manuscript is well-presented and timely, the experiments are thoughtfully designed and executed, and the results are clear. My opinion is that the manuscript is generally well-suited for publication for Nature Communications.

We were happy to learn that the Reviewer thought that our study was clear, thoughtfully designed and well-suited for publication in Nature Communication.

To fully support the authors' claims that TAF10 knockout induces TAF8 protein degradation (Fig. 4), a simple proteasomal inhibition experiment (e.g. with MG132) should be performed to see if TAF8 protein levels are stabilized. Otherwise, the decreased protein levels could be primarily attributed to the decreased mRNA levels. Similarly, since the authors have expression constructs of TAF10, it would be nice to see a rescue re-expression experiment in the TAF10 knockout cells to demonstrate that the regulation of TAF8 levels is acute (e.g. dependent on cotranslational assembly).

We carried out the proteasomal inhibition experiment, as suggested by the reviewer. We tried several timepoints (1, 3, 9, 16, 22h) for MG132 treatment. We detected a weak stabilisation of TAF8 protein level with 1 and 3h of MG132 treatment. However, we could not continue the experiment with other time points as the treatment caused massive cell death. Thus, the rescue experiment with in the *Taf10* knockout ES cells in the presence of MG132 turned out infeasible.

Thus, as also suggested by the Reviewer, we carried out a rescue experiment. To this end we turned to our mouse F9 cell system, where the two alleles of the *Taf10* gene have been knocked out and the cells survive due to the doxycycline-induced expression of the human TAF10 protein from a stably incorporated reporter construct (Metzger et al. 1999, EMBO J). We cultured the cells in the absence of doxycycline for 5 days which led to an almost complete loss TAF10 protein. In agreement with our mouse knock out data, TAF10 protein ablation was accompanied by a significant loss of TAF8 protein level. Importantly when the cells were rescued by the addition of doxycycline for either one or two days, not only were TAF10 protein levels restored, but also TAF8 protein levels became detectable (see new Figure 4e). This experiment (described on page 7 and presented as a new Figure 4e) clearly demonstrates, as suggested by the Reviewer, that TAF8 levels are regulated by co-translational assembly.

Minor comments:

1. The title does not make sense to me. "Co-translation drives the assembly..." implicates that two subunits are translated at the same time, but the authors do not directly demonstrate this. Instead, one completely synthesized subunit can interact with another that is being translated. Changing the title to "Co-translational assembly of mammalian nuclear multisubunit complexes", or something similar, seems more accurate.

We changed the title as suggested by the Reviewer.

2. In the abstract, the authors cite "protein translation". I suggest changing this to "protein synthesis" (mRNAs are translated, proteins are not).

We apologize for this mistake. We modified the Abstract as suggested.

REVIEWERS' COMMENTS:

Reviewer #1 (Remarks to the Author):

The authors of this manuscript have done a great job of responding to the initial reviews of all three referees. This study makes substantial new and important advances in our understanding of the mechanisms that contribute to the synthesis and assembly of large, complex, multi-subunit biologically important molecular machines.

The authors made substantial editorial and experimental additions to their revised manuscript. All of these changes have significantly improved this study. It is opinion of this reviewer that this revised manuscript should be accepted for publication in Nature Communications. This work will dramatically change how we think and approach multsubunit complex assembly going forward.

P. Anthony Weil

Reviewer #2 (Remarks to the Author):

I am satisfied with the response to my request about two-color FISH of TAF6 and TAF9. But I am not convinced by the argument that endogenous TAF10-IF on TAF10 mRNA or endogenous TAF8 mRNA. The authors argue that the transient nature of nascent TAF10 protein precludes their detection on TAF10 mRNA. The same should also be true for transient nature of nascent TAF8 proteins. In addition, the percent of colocalized TAF10 IF on TAF8 mRNA should not depend on the expression of TAF8, but rather on the level of TAF10 protein in cytoplasm.

Reviewer #3 (Remarks to the Author):

This is a beautifully designed and executed study analyzing the assembly of multi-component protein complexes. The authors present clear evidence supporting the cotranslational assembly of several nuclear complexes, and that these interactions are required for mRNA and protein stability of

specific subunits. These data will be of wide interest to the protein biosynthesis and posttranscriptional gene regulation fields.

The authors adequately addressed my original concerns in their revision. My only minor residual comment is that this particular study does not present evidence to rigorously demonstrate that failing to cotranslationally assemble directly causes protein and mRNA degradation (vs. transcriptional or translational responses). Doing so requires observing protein stabilization when disrupting protein degradation, measuring the half-lives of the mRNA in steady state conditions vs. conditions in which assembly is not possible, and/or demonstrating that mRNA is stabilized when translation is inhibited - a hallmark of translational arrest-induced mRNA degradation. I believe these experiments are beyond the scope of the study, but that the text should more accurately portray which aspects of the suggested model remain speculative relative to the data presented (this was not so clear in lines 188-191 and 308-314). For example, although the model of translational pausing causing mRNA destabilization is an attractive explanation for what is happening and the additional data in the new Fig. 4e is very convincing, these results could still be reflecting a primarily transcriptional instead of posttranscriptional response since it was done on the timescale of days instead of hours. Mentioning these other possibilities will help to avoid misleading readers who are not familiar with these types of experiments. Other than this minor comment, I fully support publication.

REVIEWERS' COMMENTS:

Reviewer #2 (Remarks to the Author):

I am satisfied with the response to my request about two-color FISH of TAF6 and TAF9.

We were happy to learn that the Reviewer is satisfied with the two-color TAF6-TAF9 FISH.

But I am not convinced by the argument that endogenous TAF10-IF on TAF10 mRNA or endogenous TAF8 mRNA. The authors argue that the transient nature of nascent TAF10 protein precludes their detection on TAF10 mRNA. The same should also be true for transient nature of nascent TAF8 proteins.

We apologize, but there may have been a misunderstanding. We did not claim that the transient nature of nascent TAF10 protein on TAF10 mRNA is the sole reason that precludes its detection. It is primarily due to the fact that we do not have a TAF10 antibody that binds to the N-terminal region of the TAF10 protein, which is absolutely necessary to detect nascent TAF10 protein on *TAF10* mRNA. The anti-TAF10 antibody that we used in Fig 5 binds to the middle region of TAF10 protein and is not useful to detect nascent TAF10 protein. Unfortunately, we do not have a good anti-TAF10 antibody that would bind to the N-terminal end of the nascent endogenous TAF10 protein.

In addition, Panasenko et al (NSMB, 2019) published recently that if two protein partners are assembling co-translationally, the translating nascent protein partner undergoes a ribosome pause following translation of its interaction domain (shown by ribosome profiling data) until its partner comes and binds to it and this phenomenon was suggested by Panasenko et al (2019) to be conserved from yeast to human cells. This would mean that in our case, nascent TAF8 translation would pause following translation of its histone fold interaction domain and translation would only continue after TAF10 would bind to it. Thus, it is conceivable that this pause would make it somewhat easier for us to detect the co-localization of TAF10 protein on *TAF8* mRNA than TAF10 protein on the *TAF10* mRNA.

In addition, the percent of colocalized TAF10 IF on TAF8 mRNA should not depend on the expression of TAF8, but rather on the level of TAF10 protein in cytoplasm.

TAF10 is a part of two multi-subunit complexes: TFIID and SAGA complexes. Our data points towards the fact that TAF10 co-translationally assembles with TAF8, which is a subunit of TFIID complex. The total population of TAF10 in the cell is divided among two complexes. Thus, increasing the number of *TAF8* molecules (rather than *TAF10* molecules) would enable us to better visualize the co-translational assembly of TAF10 and *TAF8*. Note however, that even without *TAF8* mRNA overexpression we observe TAF10 protein co-localization with endogenous *TAF8*, but with a very low frequency (Supplementary Figure 4c).

Reviewer #3 (Remarks to the Author):

This is a beautifully designed and executed study analyzing the assembly of multi-component protein complexes. The authors present clear evidence supporting the cotranslational assembly of several nuclear complexes, and that these interactions are required for mRNA and protein stability of specific subunits. These data will be of wide interest to the protein biosynthesis and posttranscriptional gene regulation fields.

The authors adequately addressed my original concerns in their revision. My only minor residual comment is that this particular study does not present evidence to rigorously demonstrate that failing to cotranslationally assemble directly causes protein and mRNA degradation (vs. transcriptional or translational responses). Doing so requires observing protein stabilization when disrupting protein degradation, measuring the half-lives of the mRNA in steady state conditions vs. conditions in which assembly is not possible, and/or demonstrating that mRNA is stabilized when translation is inhibited - a hallmark of translational arrest-induced mRNA degradation. I believe these experiments are beyond the scope of the study, but that the text should more accurately portray which aspects of the suggested model remain speculative relative to the data presented (this was not so clear in lines 188-191 and 308-314). For example, although the model of translational pausing causing mRNA destabilization is an attractive explanation for what is happening and the additional data in the new Fig. 4e is very convincing, these results could still be reflecting a primarily transcriptional instead of posttranscriptional response since it was done on the timescale of days instead of hours. Mentioning these other possibilities will help to avoid misleading readers who are not familiar with these types of experiments. Other than this minor comment, I fully support publication.

We were happy to learn that the Reviewer thought that our study was “beautifully designed and executed” and s/he is satisfied with our revision.

We apologize for not explaining properly all the possible models. As required we have changed the text following the suggestions of the Reviewer. Consequently, the lines pointed out by the Reviewer (on page 7 and 12) have been changed to acknowledge the suggested different possibilities.